

# Wind tunnel tests with combined pitch and free-floating flap control: Data-driven iterative feedforward controller tuning

S. T. Navalkar[1], L. O. Bernhammer[2], J. Sodja[3], E. van Solingen[1], G. A. M. van Kuik[2], and
J. W. van Wingerden[1]

[1]Delft Center for Systems and Control, Faculty of Mechanical, Maritime and Materials Engineering, Delft University of
Technology, 2628 CD Delft, the Netherlands.
[2]Wind Energy Group, Faculty of Aerospace Engineering, Delft University of Technology, 2629 HS Delft, the Netherlands.
[3]Aerospace Structures and Computational Mechanics, Faculty of Aerospace Engineering, Delft University of Technology,
2629 HS Delft, the Netherlands.

*Correspondence to:* Sachin Navalkar (S.T.Navalkar@tudelft.nl)

**Abstract.** Wind turbine load alleviation has traditionally been addressed in literature using either full-span pitch control, which
has limited bandwidth, or trailing-edge flap control, which typically shows low control authority due to actuation constraints.
This paper combines both methods, and demonstrates the feasibility and advantages of such a combined control strategy
on a scaled prototype in a series of wind tunnel tests. The pitchable blades of the test turbine are instrumented with free-
floating flaps close to the tip, designed such that they aerodynamically magnify the low stroke of high-bandwidth actuators.
The additional degree of freedom leads to aeroelastic coupling with the blade flexible modes. The inertia of the flaps was tuned
such that instability occurs just beyond the operational envelope of the wind turbine; the system can however be stabilised
using collocated closed-loop control. A feedforward controller is shown capable of significant reduction of the deterministic
loads of the turbine. Iterative feedforward tuning, in combination with a stabilising feedback controller, is used to optimise the
controller online in an automated manner, to maximise load reduction. Since the system is non-linear, the controller gains vary
with wind speed; this paper also shows that iterative feedforward tuning is capable of generating the optimal gain schedule
online.

## 1 Introduction

The increasing size and flexibility of wind turbines demand that attention be devoted towards the active and passive control
of rotor loads in order to limit the costs related to both the construction as well as maintenance of the turbine blades and the
support structure. One of the most interesting and readily accessible methods of blade load control is Individual Pitch Control
(IPC) Bossanyi (2003), whereby each blade is pitched along its longitudinal axis independently to counteract the variation in
wind loading. Numerous references can be found in literature which prove the efficacy of IPC in load control of wind turbines,
both in the simulation environment Selvam et al. (2009), Bottasso et al. (2013), as well as experimentally Bossanyi et al. (2013).
In these references, with ordinary levels of turbulence, it has been observed that IPC can achieve upto 30% reduction in the
standard deviation of blade loads. Previous experimental studies conducted by the authors Navalkar et al. (2015) show that in



a controlled, wind tunnel environment, wind turbine blade load reductions upto 70% can be reached, since the blade loading under these circumstances is almost entirely deterministic. However, in all the references mentioned, the target of IPC has been to reduce low-frequency loads, primarily around the 1P (rotor frequency). While IPC can herewith address a large part of load spectrum, the emphasis on low frequencies is also a product of the low bandwidth that can be achieved with the full-span pitch

control, which involves actuation of the large torsional inertia of the blades around their axes. As expected, IPC also leads to a substantial increase in pitch activity.

In an effort to reduce pitch actuator duty, target higher frequencies in the load spectrum, and address localised disturbances in the wind loading, recent literature has explored the concept of the 'smart' rotor Lackner and Van Kuik (2010): a rotor where the blades are instrumented with sensors and flow-modifying actuators at various radial locations. Reviews of such rotors Barlas

and Van Kuik (2010) Bernhammer et al. (2012) invariably conclude that trailing-edge flaps give the best control authority for load alleviation. The load alleviation potential has been demonstrated in simulations Andersen et al. (2006) Bernhammer et al. (Article in Press) and experimentally in a wind tunnel Van Wingerden et al. (2011). Further, field tests of this concept have also been conducted Castaignet et al. (2013), although such a system is still not considered mature enough for incorporation in a commercial wind turbine. While these tests used conventional actuators, many references recommend the useage of smart

actuators, such as piezoelectrics, in order to enhance bandwidth and achieve a high power-to-weight ratio. Such actuators unfortunately show low stroke and hence reduced control authority.

The concept of the free-floating flap Heinze and Karpel (2006) combines a trailing-edge flap that is free to rotate about its axis, with a small tab located on the flap, that can be actuated at a high speed to dynamically change flap camber. This concept was developed specifically to take advantage of aerodynamic levering to increase the low stroke of smart actuators. For a fixed

wing instrumented with such a free-floating flap, it was experimentally shown Bernhammer et al. (2013) that it is possible to achieve enhanced control authority. Further, this study also demonstrated that such a flap could be completely autonomous in terms of energy consumption, and can be used as a plug-and-play device. This modularity shows promise for the construction and maintenance of future smart blade. However, this concept has not yet been demonstrated experimentally on a wind turbine.

Numerical and experimental investigations of the free-floating flap concept Heinze and Karpel (2006) Bernhammer et al.

(2013) have shown that the additional degree of freedom adds a rigid-body mode to the system, the dynamics of which are strongly dependent on the total air speed at operation. Aeroelastic coupling of this mode with the flexible blade mode induces flutter at low wind speeds, an instability that can lead to dangerously high vibrations and even structural failure. However, it has also been shown in the references that closed-loop control of the tab can ensure safe operation of a fixed wing, well into the unstable regime. A pitchable wind turbine blade instrumented with free-floating flaps thus poses several control challenges.

Firstly, the nature of the flap implies that its dynamic response is not constant but varies strongly with the wind speed. Such a system cannot be described by a linear time-invariant (LTI) state-space realisation, but can possibly be expressed as a linear-parameter-varying (LPV) system, where the time-varying parameter depends on the wind speed. Further, the presence of a stabilising closed-loop controller is mandatory. Finally, the uncertainties in flow and structure modelling suggest that a data-driven controller may be able to achieve greater optimality of performance without the excessive conservatism of true robust

design.





Data-driven control of wind turbine loads has been demonstrated experimentally in Navalkar et al. (2015), where online recursive system identification was combined with online controller synthesis for minimising the periodic turbine loads. However, such a controller would be required to retune itself at every instant the ambient wind conditions change. An alternative methodology for the data-driven alleviation of wind loads has been described in Navalkar and Van Wingerden (2015) which

employs the iterative feedback tuning (IFT) Hjalmarsson (2002) methodology to tune the gains of a fixed structure controller, hereby optimising a (convex) performance criterion. While IFT controllers have been used in the industry, they have typically been implemented to converge to linear time invariant controller structures Gevers (2002). The use of IFT for tuning the gains of time-varying controllers, as required for the current application, has been described in literature Navalkar and Van Wingerden (2015) but not yet demonstrated in practice.

The contribution of this paper is thus threefold: firstly, scaled wind turbine blades instrumented with outboard free-floating flaps are designed and manufactured for wind tunnel testing. Secondly, the load alleviation potential of the free-floating flaps in combination with individual pitch control is demonstrated for the first time in an experimental sense. The load alleviation potential is investigated in both the stable and unstable (post-flutter) modes of operation, and the importance of collocated control will be highlighted. Finally, the setup will serve as a test bench for a novel iterative feedback tuning algorithm that

automatically tunes a controller gain schedule for load alleviation in real-time variable wind speed operation.

The remainder of the paper is organised as follows: Section II describes the design and manufacturing process for the wind turbine blades with free-floating flaps. Section III gives a brief description of the testing environment. The aeroelastic behaviour of the blades is studied in Section IV. The control algorithm used for load alleviation is formulated in Section V. The results of the testing are laid out in Section VI, and conclusions are drawn from these results in the final section.

## 2   Blade design and manufacturing

The design of the blades formed the most important part of the design process of the scaled turbine, since it had to form a reasonable approximation of a full-scale wind turbine blade while adhering to the constraints set by the wind tunnel capabilities. The comparison of the scaled turbine with the Innwind reference turbine Bak (2013) is given in Table 1. The primary scaling that was aimed to be achieved was maintaining the ratio of blade first eigenfrequency to rotor speed (1P), as is done in Van

Wingerden et al. (2011). This ratio is 3.5 for the reference turbine.

### 2.1   Blade design

The overall aerodynamic and structural design of the blades follows the procedure described in Van Wingerden et al. (2011), since the blades were designed for similar wind tunnel testing conditions. However, as the wind tunnel experiments will also

incorporate blade pitch control, the torsional inertia of the blades was reduced by scaling down the root chord by 30%. The root chord thus measures 200 mm, tapering to a tip chord value of 120 mm over a blade length of 750 mm, with a total blade twist of 12°.



**Table 1.** Parameter Comparison of the Scaled Turbine

|  | Reference turbine Bak (2013) | Scaled turbine |
| --- | --- | --- |
| Rotor diameter (m) | 178.3 | 2 |
| Rated wind speed (m/s) | 11.4 | 4.5 |
| Tip speed ratio (-) | 7.86 | 5.35 |
| Rated rotational speed (rpm) | 9.6 | 230 |
| Fore-aft tower mode (Hz) | 0.25 | 4.5 |
| First flapwise mode (Hz) | 0.56 | 14.4 |
| Ratio 1st blade freq. to 1P (-) | 3.5 | 3.75 |

Out of structural considerations, it was deemed necessary to minimise the weight of the blades, while ensuring adequate structural integrity to withstand the centrifugal and out-of-plane loading that the blade will be subject to during operation. An accurate aerodynamic shape of the blade was ensured by 3d printing the blade and then further reinforced with unidirectional carbon fibre spar caps, as shown in Figure 3. Small wind turbine blades have previously been manufactured in this manner by the University of Stuttgart Bauer et al. (2014), and a comparison of different additive manufacturing techniques can be found in Karutz (2015). These references specifically investigate 3D printing of blades in a set of sections that are bonded together. In order to avoid solid plastic-plastic joints, it was decided that the blades in the current case would be printed in one piece.

Three different materials (ABS M30, PC-ABS and nylon), that can be used for 3d printing, were evaluated qua their ability to bond with carbon fibre. For each material, a rectangular sample of size 200 mm x 30 mm, of thickness 3 mm, was 3d printed. Subsequently, each sample was bonded on the top and bottom with a single layer of unidirectional carbon fibres of thickness 0.14 mm, impregnated with epoxy resin. A four-point bending test to failure was then conducted with each of the samples. The distance between the supports was 140 mm, while the points of force application were 40 mm apart. The results of the test can be seen in Figures 1 and 2. In Figure 1, the behaviour to failure in bending can be observed. For small loads, the response is linear. At higher loads, small kinks can be observed in each of the curves, these physically represent the snapping of individual carbon fibres in compression. Finally, there is a large drop in strength when delamination occurs in the materials ABS M30 and nylon. For the material PC-ABS, brittle fracture occurs before delamination, as such the bond between this material and the carbon fibre spar is the best for this material. Further, it also holds its strength over a larger range of deformation than the other materials. Since PC-ABS also shows good mechanical workability, the choice was made to 3d print the scaled blade using this material.

The blade was printed as a 3 mm-thick shell, with an internal spar structure, using stereolithography techniques. In order to add structural stiffness to the blades, a spanwise slot was engraved at the spar cap location on both the top and bottom of the blade. This slot was filled with a 0.14 mm thick layer of unidirectional carbon fibre tow impregnated with epoxy resin. The slot was then aerodynamically faired using crushed glass fibre epoxy filler, which was then sanded down for a smooth finish. A CAD model of the blade and a photograph of the finished blade are shown in Figures 3 and 4. The CAD software Solidworks




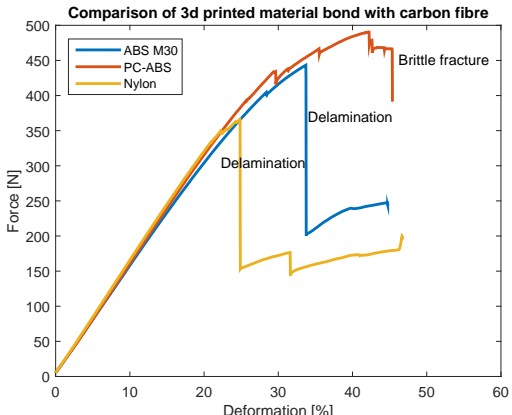

**Figure 1.** Structural behaviour of the bond between 3d printed substrate and carbon fibre spar

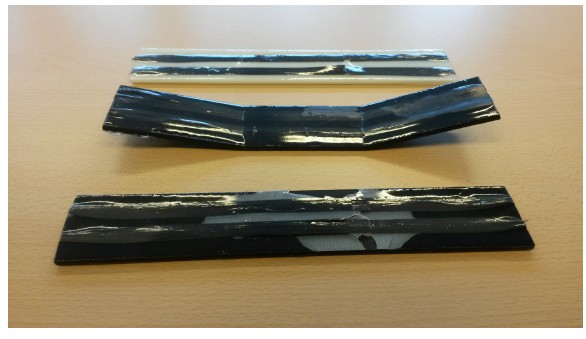

**Figure 2.** 3d printed samples post fracture. Top: ABS M30, Middle: PC-ABS, Bottom: Nylon

was used for designing the blade, with the blade material considered to be homogeneous and isotropic. The metal connection to the hub and the carbon fibre are modelled to be bonded to the blade ideally such that delamination is not possible. An ultimate loading case is simulated for a wind speed of 10 m/s, rotor speed of 400 rpm and a thrust coefficient of 1. For this extreme case, the stresses in the plastic material are calculated to be less than the flexural strength of the material by a factor of safety of 1.3.

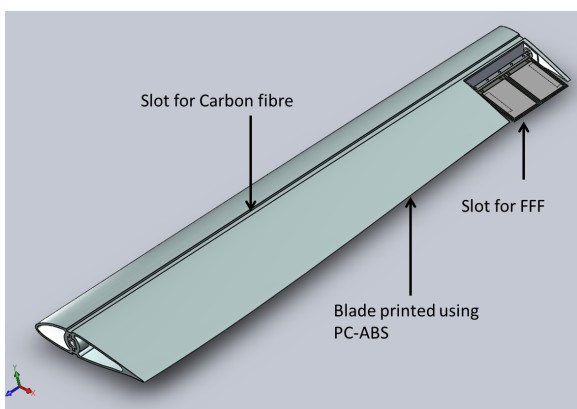

**Figure 3.** Blade CAD model

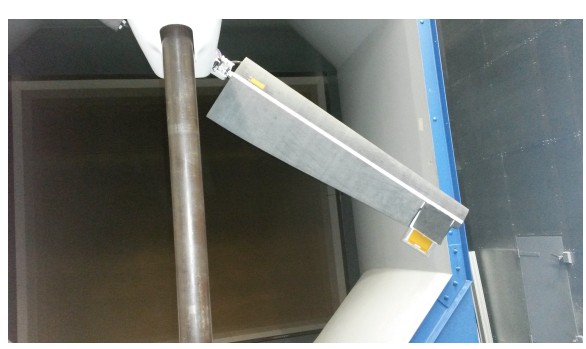

**Figure 4.** Photograph of Blade

5   The designed static force-deflection curve, compared with the measured structural behaviour, is seen in Figure 5. It can be directly observed that the blade designed using CAD software is 30% stiffer than the actual manufactured blade. Two reasons can be postulated for this: firstly, the bonding between the various parts of the assembly is not perfect, leading to flexibility and play in the actual blade. Further, the 3d printing layers are perpendicular to the longitudinal axis of the blade, thereby weakening the blade in a flapwise direction in an inhomogeneous manner not captured by the isotropic FEM model. It is,




however, interesting to note that the predicted stiffening effect of the carbon fibre layer is nearly identical. The tip deflection was calculated to be 17.2% lower with carbon fibre spars, while it was measured to be 16.6% lower after stiffening. A flexible mode analysis of the blade yields its first natural frequency as 16.43 Hz, as depicted in Figure 6. Since the measured stiffness is 30% lower than the calculated stiffness, the first eigenfrequency of the blade is also lower than its calculated value, and is measured to be 14.4 Hz, thereby achieving the desired frequency scaling.

Post manufacture, the blades are instrumented with piezoelectric strain sensors on the top and bottom, at the root of each blade. These sensors provide a measure of the blade loads that are sought to be minimised.

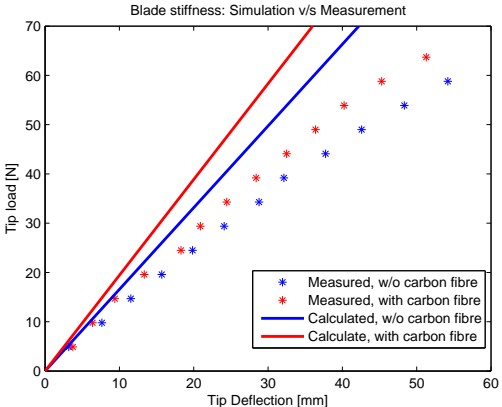

**Figure 5.** Calculated stiffness characteristics compared with measured stiffness characteristics

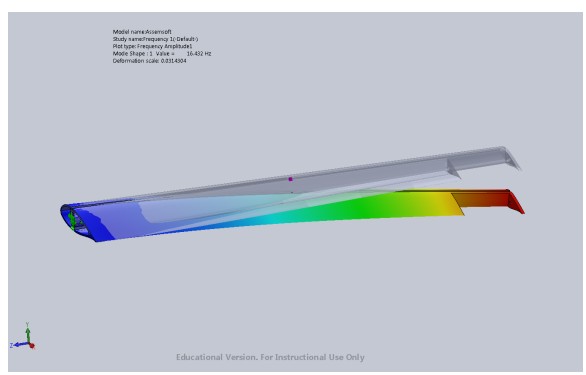

**Figure 6.** Calculated first eigenmode of blade
Frequency 16.43 Hz

## 2.2 Free-floating flap (FFF) design

The CAD design of the free-floating flaps is depicted in Figure 7. The leading edge is a continuation of the inboard portion of the blade, including the slot meant for carbon fibre stiffening. The hinge axis of the flap is mounted using bearings on an aluminium bracket just behind the spar; apart from the negligible bearing friction, it is entirely free to rotate. A T-section is connected to this axle, such that its interference with the mounting bracket provides limit stops for the rotation of the flap. The flap can hereby rotate freely through a maximum upward and downward deflection angle of 30°.

A metal plate (spring steel) of thickness 0.2 mm is sandwiched between the axle and the T-section. Two piezobenders (Macrofibre composite, type M8557-P1) are affixed rigidly to the top and the bottom, respectively, of this metal plate. The benders are electrically connected together in an antiparallel manner such that their piezoelectric effect reinforces each other and they produce the same magnitude but an opposite direction of strain in the substrate. A maximum voltage of +/- 500 V can be applied to the benders in order to emulate the behaviour of the trim tab from Heinze and Karpel (2006) and Bernhammer et al. (2013). Finally,an appropriate aerodynamic shape of the flap was achieved by embedding the instrumented metal plate into a highly compliant foam which was shaped according to the aerofoil geometry. The entire flap, from the angle-limiting





T-section to the foam spacers, is covered with a fairing shroud. A contactless angle encoder is embedded into the tip section, which provides feedback on the flap angular position.

This configuration causes a step change in the chordwise profile just aft of the spar, that produces undesirable aerodynamic behaviour, which is a well-known trade-off against the increase in the deformability of the trailing edge. In this experiment, to

achieve a proof of concept for free-floating flaps, aerodynamic accuracy is sacrificed for control authority in the design of the flap.

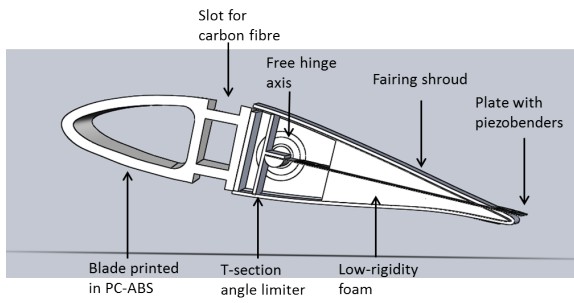

**Figure 7.** Flap cross-section: trim tabs replaced by chordwise piezobenders

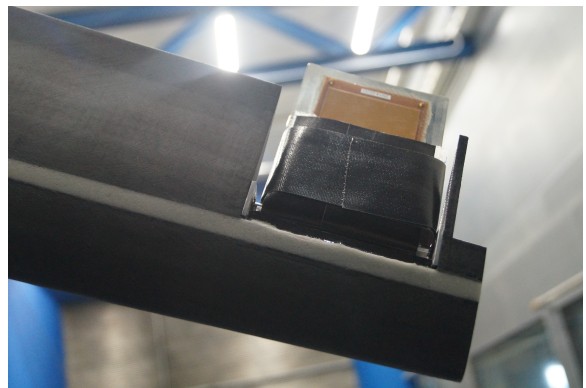

**Figure 8.** Photograph of flap

## 3   Aeroelastic blade analysis

While in the previous section it was ensured that the behaviour of the blade under the ultimate static stuctural load was acceptable, an aeroelastic analysis is required to determine the change in its structural response with increasing wind speed.

Both Bernhammer et al. (2013) and Bernhammer et al. (2015) note that the free-floating flap adds an additional rigid-body degree of freedom, a structural mode which is strongly dependent on the operating air speed. This mode is expected to couple with the first flexible mode of the system, giving rise to a low wind speed form of flutter.

In order to analyse the aeroelastic behaviour of the blade instrumented with a free-floating flap, the blade is modelled in MSC/NASTRAN as a cantilever beam of non-uniform cross-section (CBAR elements). The various cross-sections of the

modelled beam were taken at ten equidistant spanwise stations along the blade. Each element is rigidly connected to a flat-plate aerodynamic panel of the corresponding chordwise length. The flap is modelled in a similar manner. First, a structural modal analysis of the blade is carried out, at zero wind speed. The calculated modes of the blade are given in Table 2. A few of the most important modes have been visualised in Figures 9 to 12. It is most interesting to note that the lowest flexible mode is the flapwise mode, with modal frequency 19.4 Hz. The corresponding frequency predicted by Solidworks is 16.43 Hz. The

difference in the results is possibly due to the difference in modelling: Solidworks uses a full three-dimensional blade analysis, while the NASTRAN analysis uses one-dimensional beam elements. This is the mode most likely to couple unstably with the rigid body flap mode. The blade is significantly stiffer in both the lead-lag and torsional directions; these modes are hence



unaffected by aerodynamic coupling. An actual turbine blade Bak (2013) is relatively softer in these directions, however even for such a blade, the flapwise mode is the most relevant one for load analysis and also possesses the lowest frequency. The current scaled blade design, with high lead-lag and torsional stiffness, allows us to study the low-speed flutter phenomenon with limited complexity.

**Table 2.** Structural modes of the blade at zero total air speed

| Mode description | Modal frequency | Mode description | Modal frequency |
|---|---|---|---|
| Rigid-body flap mode | 0 Hz | 1$^{st}$ Flapwise mode | 19.44 Hz |
| 1$^{st}$ Lead-lag mode | 76.67 Hz | 2$^{nd}$ Flapwise mode | 87.88 Hz |
| 3$^{rd}$ Flapwise mode | 223.9 Hz | 2$^{nd}$ Lead-lag mode | 291.3 Hz |
| 1$^{st}$ Torsional mode | 361.6 Hz | 4$^{th}$ Flapwise mode | 449.6 Hz |

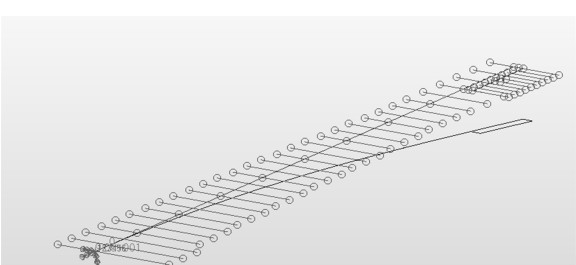

**Figure 9.** First flapwise mode: Mode 2, Frequency 19.44 Hz

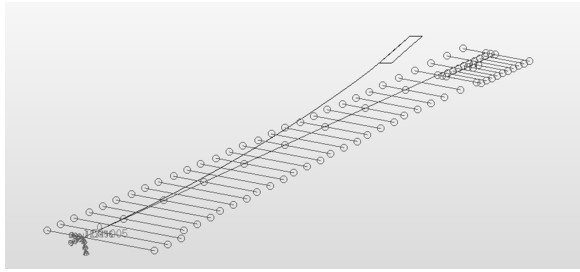

**Figure 10.** First lead-lag mode: Mode 3, Frequency 76.67 Hz

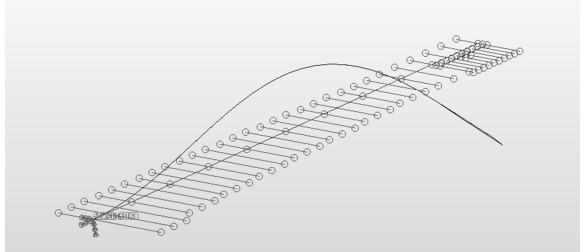

**Figure 11.** Second flapwise mode: Mode 4, Frequency 87.88 Hz

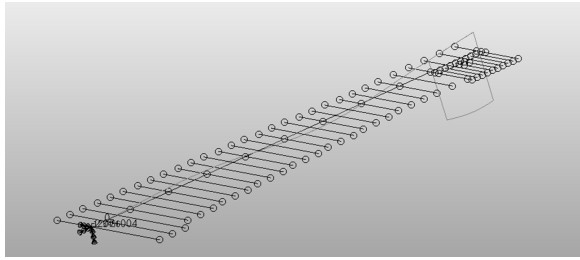

**Figure 12.** First torsional mode: Mode 7, Frequency 361.6 Hz

The low-speed flutter phenomenon, as predicted by NASTRAN, can be seen in Figures 13 and 14. It can be directly observed that the frequency of the rigid body flap mode rises linearly with wind speed. Due to coupling of this mode with the first blade



flexible mode, the blade mode becomes unstable at the onset of flutter at a total air speed of 36 m/s, which corresponds to a turbine rotor speed of 340 rpm, thus at a speed beyond the designed operational speed of the wind turbine (230 rpm).[1]

This aeroelastic analysis served as a guideline for the design of the blades and for identifying the range of operation permissible in the experiments described in the sequel. Experimentally, it was observed that the onset of flutter occurred at 315
rpm. However, since this mode involves exponentially diverging vibrations in the blades, which cannot be physically limited, open-loop experiments in this unstable regime were not conducted out of safety considerations.

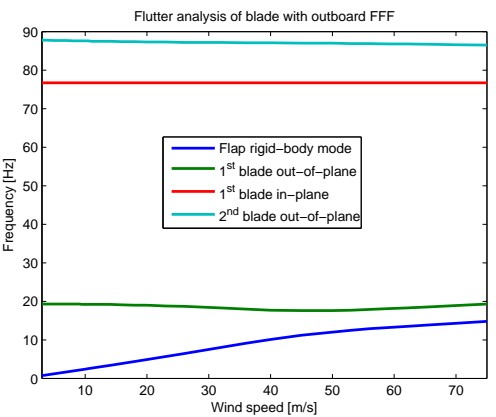

**Figure 13.** Variation in modal frequency with total incident air speed.

**Figure 14.** Variation in modal damping with total incident air speed.

## 4   Testing Environment

The blades designed and analysed as above were mounted on the test turbine setup used previously in Navalkar et al. (2015). As described in this reference, the blades are connected to the hub through pitch servomotors.
The hub is mounted on a shaft supported by two main bearings located in the nacelle. The electrical connections of the hub are transferred to the stationary part of the wind turbine via slip rings, rated at 500 V, which is also the maximum voltage that can be fed to the piezobenders located outboard on the blades. Further, the shaft is instrumented with a torque transducer and speed encoder, and connected mechanically with the generator. The turbine is direct-drive; the rotor speed is the same as the generator speed. The generator is in turn connected electrically in series with an adjustable dump load amenable to resistance
control. Thus, in principle this setup can also provide torque control. However, in this series of tests, the resistance of the dump load is kept constant. This implies that the wind turbine is in constant torque operation, and its rotor speed rises linearly with the incoming wind speed.

---

[1]In principle, a higher (pre-flutter) rotor operational speed of upto 300 rpm could have been chosen; the tip speed ratio is in both cases virtually identical. However, the speed of 230 rpm gives the best ratio of forcing frequency to blade eigenfrequency. Further, the current tower design yields a very lightly damped tower torsional mode at 280 rpm, which is deemed necessary to avoid out of practical considerations



The nacelle is connected rigidly to the top of a tower, mounted on bearings on its base. The tower (and hence the entire wind turbine) can yaw freely around its base. For this set of experiments, the tower is kept fixed such that the plane of the rotor is always perpendicular to the incoming wind speed.

The entire assembly is mounted inside the Open Jet Facility of the Delft University of Technology, which is an open jet wind

5   tunnel of 6 m test cross-section and 2.85 m equivalent open jet diameter. A photograph of the turbine can be seen in Figure 15. While wind speeds up to 35 m/s can be achieved in this wind tunnel, the operation of the wind turbine under the current settings requires no more than 6 m/s, with a rated wind speed of 4.5 m/s (and thus a tip speed ratio of 5.35).

Data acquisition and online control is furnished at a sampling frequency of 2 kHz by a real-time PC, on which the controller is compiled using Matlab-Simulink xPCTarget. There are two primary sensing elements: the load sensors at the blade roots

10  and the free-floating flap angle sensors. Further, there are two primary actuators: the piezobenders on the flaps and the pitch motors. The objective of the experiments is to use these sensing and actuating elements to achieve load control of the scaled wind turbine.

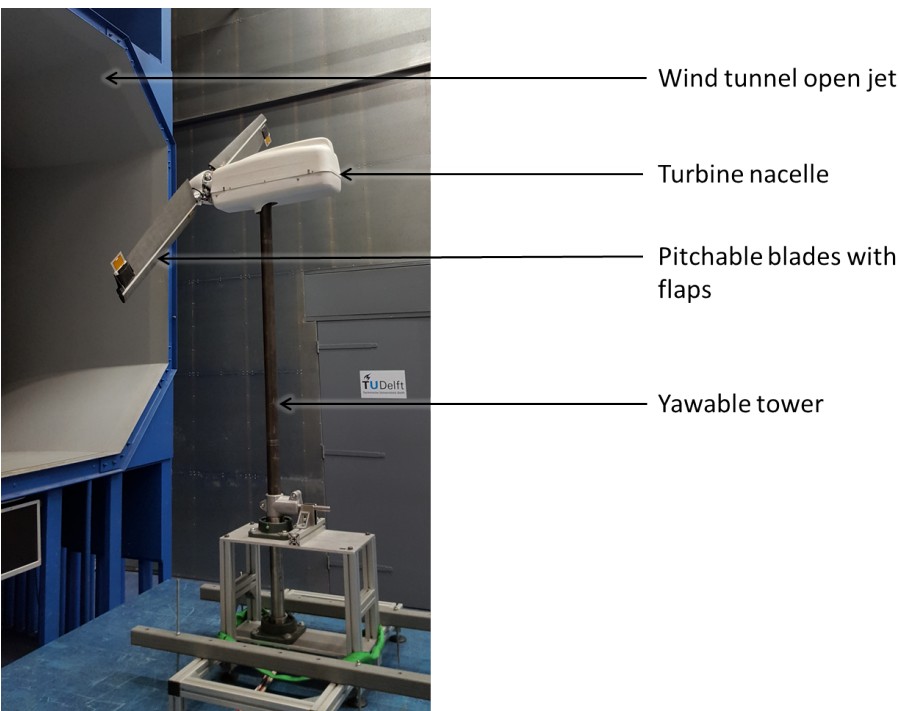

**Figure 15.** Photograph of the assembled turbine with pitch and flap control.



## 5 Iterative Feedforward Tuning for Combined Pitch and Flap Control

For a wind turbine in the field, the blade loads arise mainly out of wind shear, tower shadow, turbulence and its rotational sampling. As such, the blade load spectrum for a typical turbine shows dominant peaks at the rotor speed (1P) and its harmonics: for a two-bladed turbine at 2P, 4P, and so on, while for a three-bladed machine at 3P, 6P etc. The presence of turbulence broadens

these peaks and adds energy in the high frequency region of the spectrum.

In the wind tunnel environment, the levels of turbulence are low. The main cause of the blade loads is the tower passage, which leads to sharp peaks at 1P and its harmonics. The objective of the experiments is to demonstrate that these peaks can be attenuated by pitch and flap control, which by extension implies that a significant portion of the load spectrum of an in-field turbine can be addressed by these actuation and control methods.

For achieving load control, Iterative Feedforward Tuning (IFT) of the pitch and flap controllers will be implemented. This technique specifically targets deterministic disturbances, as seen in the blade load spectrum of the turbine, with minimal control action. As long as there exists a nominally stabilising controller in the loop to avoid the unstable flutter region, the controllers tuned using IFT will not render the plant unstable. Further, IFT ensures that data-driven tuning of the controllers makes them converge to an optimal control action over a number of iterations.

The optimal controller parameters depend strongly on the incoming wind speed and hence demand a linear parameter-varying (LPV) controller. LPV controller tuning using IFT has been explored and shown to work in the simulation environment Navalkar and Van Wingerden (2015). However, the computational burden and number of experiments required for tuning imply that this methodology is required to be modified to meet the demands of real-time control in the wind tunnel. Hence, a quasi-LPV approach will be followed in this section. Accordingly, it is assumed that the plant is still LPV in nature, how-

ever the wind speed varies slowly in the wind tunnel and can be approximated as constant for the duration of each set of IFT experiments.

Firstly, the notation for this section will be introduced, then the three IFT experiments will be described, and finally, the method for creating a gain schedule for controller wind speed adaptivity is described.

### 5.1 Preliminaries and Notation

An LPV formulation is set up for describing the wind turbine system, either in open-loop, or in closed-loop with a nominally stabilising controller that allows the system to be operated in a post-flutter regime:

$$x_{k+1} = A_k x_k + B_k u_k, \tag{1}$$

$$y_k = C_k x_k + D_k u_k + v_k. \tag{2}$$

In these equations, $x_k \in \mathbb{R}^{n_x}$ is the state vector of unknown size, $u_k \in \mathbb{R}^4$ are the control inputs, including the two pitch signals

and the two flap signals, while $y_k \in \mathbb{R}^2$ are the blade load signals as measured by the sensors located at the roots of the two blades. The signal $v_k \in \mathbb{R}^2$ is the external forcing signal, produced in this case by tower passage; it is a superposition of a periodic signal with zero mean white noise. The state-space matrices $A_k$, $B_k$, $C_k$ and $D_k$ are considered unknown and have



the appropriate dimensions. These matrices, as well as the disturbance $v_k$, are considered to be affine functions of the ambient wind speed, $V_k \in \mathbb{R}$:

$$A_k = A^{[0]} + V_k A^{[1]}, \tag{3}$$

and similarly for the other matrices. Typically for a wind turbine controller, a measure of the wind speed is not directly available.
It is expected that advanced wind measuring sensors like the Lidar will in the future be able to provide accurate wind speed measurements Dunne et al. (2011). Meanwhile, it is possible to use the collective pitch angle in the above-rated region, and the generator speed in the below-rated region to approximate the value of ambient wind speed; in the currrent experiments the latter approach is used. In the theoretical framework, it is hence assumed that the controller possesses perfect knowledge of the wind speed.

The feedforward disturbance attenuating controller to be designed is considered to be a full LPV controller parameterised as follows:

$$\begin{aligned}
\xi_{k+1} &= A_{c,k}(\rho)\xi_k + B_{c,k}(\rho)r_k, \tag{4} \\
u_k &= C_{c,k}(\rho)\xi_k + D_{c,k}(\rho)r_k - q_k. \tag{5}
\end{aligned}$$

Here, the controller is considered to be a fixed structure controller, such as a PID controller, with state $\xi \in \mathbb{R}^{n_c}$ of fixed
dimension. The reference signal for this controller is taken to be a set of azimuth-locked basis functions, as in Navalkar and Van Wingerden (2015), thereby rendering this method a form of adaptive cyclic pitch and flap control. This form of open-loop control is depicted in the block diagram in Figure 16. For the pitch controller, these are sinusoidal functions of frequency 1P, while for the flap controller, these are sinusoidal functions of frequency 2P. Thus, the pitch and the flap control are both decoupled in the frequency domain, and are expected to strictly attenuate loads at their respective frequencies, in order to
mitigate the maximum amount of disturbance with a minimum of control effort. Thus, with two sinusoidal basis functions for each frequency 1P and 2P, the reference signal is $r_k \in \mathbb{R}^4$. The term $q_k$ refers to an auxiliary input that will be used in the IFT experiments described in the next subsection.

    The objective of IFT is to minimise the loads as measured by the load sensors; so the cost criterion is:

$$J = \frac{1}{2N}(y^T y), \tag{6}$$

where $N$ is a sufficiently long prediction horizon. For attenuating periodic loads, $N$ is taken to be a multiple of the fundamental period of these loads. The term $y \in \mathbb{R}^{2N}$ is the load signal stacked over this horizon: $y = [y_1^T, y_2^T, ..., y_N^T]^T$. The other signals are stacked in a similar manner. The key element of the IFT methodology is the optimisation of the system performance with the help of an experimentally derived performance gradient with respect to the controller parameters. This performance gradient is give by:

$$\frac{\partial J}{\partial \rho} = \frac{1}{N}\frac{\partial y^T}{\partial \rho}y. \tag{7}$$





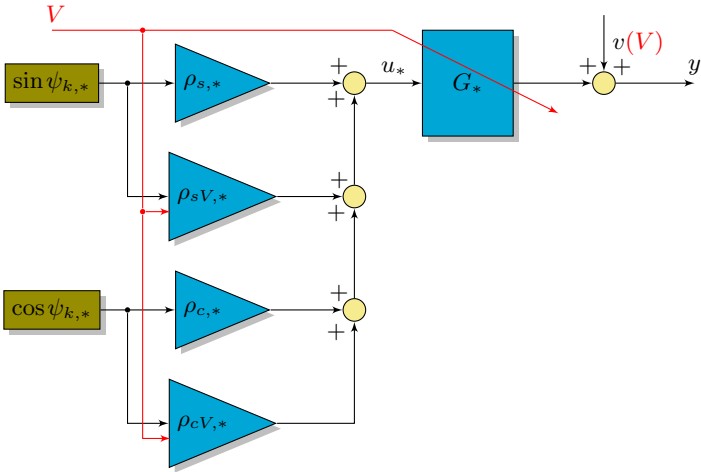

**Figure 16.** IFT implementation: wind turbine load control. $*$ stands for IPC or IFC.

Since the gradient contains stacked signals, it is more convenient to cast the system equations into a lifted format. Thus, for instance, the lifted system matrix for the wind turbine plant is given by the Toeplitz-like matrix $T \in \mathbb{R}^{2N \times 4N}$ as:

$$
T = \mathcal{T}(A_k, B_k, C_k, D_k) =
\begin{bmatrix}
D_1 & 0 & \cdots & 0 \\
C_2 B_1 & D_2 & \cdots & 0 \\
C_3 A_2 B_1 & C_3 B_2 & \cdots & 0 \\
C_4 A_3 A_2 B_1 & C_4 A_3 B_2 & \cdots & 0 \\
\vdots & \vdots & \ddots & \vdots \\
C_N A^{N-1} \ldots B_1 & C_N A^{N-1} \ldots B_2 & \cdots & D_N
\end{bmatrix}.
\tag{8}
$$

Like this lifted plant matrix $T$, a similar lifted matrix $T_c$ can be constructed for the controller. It can be observed that the system matrices are functions of the wind speed $V_k$, which is approximated to be constant for each set of IFT experiments, but changes over the course of different sets of IFT experiments (or IFT iterations). As such, for the case where the wind speed is held constant at $V_*$ for a set of IFT experiments, the system matrices of the plant and the controller are $T(V_*)$ and $T_c(V_*)$ respectively. Now, the linear time-invariant (LTI) IFT set of experiments that yield the controller gradient for the fixed wind speed $V_*$ are described next.

## 5.2 IFT Experiments

In traditional IFT Hjalmarsson (2002), designed for LTI systems, the controller parameters can be iteratively optimised by repeatedly conducting a set of three experiments for each controller parameter. If the wind speed is considered to be constant over this set of experiments, the wind turbine plant reduces to an LTI system, and the same approach can be followed for optimising controller parameters for that specific wind speed. This section recapitulates the IFT methodology with this perspective.



In the first IFT experiment, the auxiliary signal is set to zero ($q_I = 0$). It is assumed that, over the set of these experiments, the wind speed is constant at a value of $V_*$. Accordingly, the output data collected is related to the system matrices as:

$$y_I = T(V_*)T_c(V_*)r + v_I(V_*). \tag{9}$$

Considering Equation 7, in order to find the performance gradient, it is necessary to determine the gradient of the output with respect to the parameters, $\frac{\partial y}{\partial \rho}$. As such, the equation above is differentiated with respect to each controller parameter $\rho_{j_\rho}$, $j_\rho = 1, ..., n_\rho$, where $n_\rho$ is the number of controller parameters. This results in the following equality:

$$\frac{\partial y_I}{\partial \rho_{j_\rho}} = T(V_*)\frac{\partial T_c(V_*)}{\partial \rho_{j_\rho}}r. \tag{10}$$

It should be noted that in the above equation, the only unknown on the right hand side is the system matrix of the plant, $T(\mu_*)$. In order to estimate its filtering effect, the second experiment uses the following auxiliary input:

$$q_{II} = \frac{\partial T_c(V_*)}{\partial \rho_{j_\rho}}r. \tag{11}$$

Herewith, the output in the second experiment becomes:

$$y_{II} = (T(V_*) - T(V_*)\frac{\partial T_c(V_*)}{\partial \rho_{j_\rho}})r + v_{II}(V_*). \tag{12}$$

The required output gradient $\frac{\partial y_I}{\partial \rho_{j_\rho}}$ is now given by:

$$\frac{\partial y_I}{\partial \rho_{j_\rho}} = y_I - y_{II} + v_{II}(V_*) - v_I(V_*). \tag{13}$$

Since the disturbance signal $v$ is a superposition of a periodic signal and random noise, and the stacking length $N$ is a multiple of period of the noise, the term $v_{II}(V_*) - v_I(V_*)$ does not contain a periodic component, and is purely zero mean white noise. As such, the output gradient below is ergodically unbiased:

$$\frac{\partial \hat{y}_I}{\partial \rho_{j_\rho}} = y_I - y_{II}. \tag{14}$$

However, the performance gradient cannot in this case be constructed simply as:

$$\left.\frac{\partial J}{\partial \rho_{j_\rho}}\right|_{V=V_*} = \frac{1}{N}(y_I - y_{II})^T y_I. \tag{15}$$

This is because the noise in the estimate of the output gradient $\frac{\partial y_I}{\partial \rho_{j_\rho}}$ is correlated with the disturbance components in $y_I$ and the performance gradient estimate would hence be biased. So, a third experiment, replicating the deterministic conditions of the first experiment, is required to be conducted, in order to obtain the statistically uncorrelated output $y_{III}$. Finally, the performance gradient is given by:

$$\left.\frac{\partial J}{\partial \rho_{j_\rho}}\right|_{V=V_*} = \frac{1}{N}(y_I - y_{II})^T y_{III}. \tag{16}$$





With this performance gradient estimated from data, an optimisation method, such as a steepest descent method, can now be employed to obtain the optimal value of the controller parameter. It is to be noted, however, that the controller parameter derived in such a manner is optimal only for the operating wind speed. The iterations for achieving such a conditionally optimal controller parameter can be denoted by:

$$\rho_{j_\rho}^{i+1}(V_*) = \rho_{j_\rho}^i(V_*) - \gamma^i R^{-1} \frac{\partial J}{\partial \rho_{j_\rho}}\bigg|_{V=V_*}. \tag{17}$$

Here, the term $\gamma$ is an (iteration-dependent) scalar step size that can be tuned for achieving the desired convergence rate. It should be noted that a step size that is too large may lead to non-convergence. The term $R$ represents a positive definite matrix, which is identity for the steepest descent method, but can be the Hessian or an estimate of the Hessian with respect to the controller parameter for increasing the rate of convergence.

With this method, the optimal controller parameters for a specific wind speed can be iterated to. The next section details the synthesis of a gain schedule for adapting the parameters for the case with slowly varying wind speed.

### 5.3 Data-driven Gain schedule synthesis

The previous section details the manner in which, for a constant wind speed, an updated estimate of the ideal controller parameters for that wind speed can be obtained. In this section, it is assumed that, in each iteration $i$, the ideal parameters vary as an affine function of the wind speed $V^i$ in the following manner:

$$\rho_{j_\rho}^*(V^i) = \rho_{j_\rho}^{[0],*} + V^i \rho_{j_\rho}^{[1],*}. \tag{18}$$

While the above equation indicates a linear relationship between the optimal parameter and the scheduling variable $V$, this may not in practice always be the case. However, the same equation can also be extended to an arbitrarily high degree of complexity, using either polynomial or any other suitable basis functions. The choice of the number of functions depends upon the non-linearity of the scheduling, and on the signal to noise ratio achieved by the sensors, and may in practice be difficult to estimate a priori.

The objective of IFT is then to iterate to the optimal values of $\rho_{j_\rho}^{[0]}$ and $\rho_{j_\rho}^{[1]}$ based on the data inferred from the experiments described in the previous section. In the simple case of affine scheduling dependence described above, this can be achieved by recursive least squares estimation of the gain schedule. Thus, at every iteration that a pair $\rho_{j_\rho}^{i+1}$ and $V^{i+1}$ is obtained from Equation 17, recursive linear regression is used to update the gain schedule.

This procedure is repeated until the gain schedule converges to the optimal gain schedule described in Equation 18. Hereby, IFT is able to synthesise an optimal gain-scheduled combined pitch and flap controller for the case where the wind speed varies slowly. The optimal tuning of such a controller will be demonstrated experimentally on the wind tunnel setup described in the previous sections.



## 6   Results

To recapitulate, the objective of the wind tunnel experiments was to achieve blade load control for the scaled wind turbine, using full-span pitch actuation and free-floating flap control, with Iterative Feedforward Tuning for optimal performance of the load controller. It should be noted that since experiments are conducted under constant load operation, the rotor speed varies

linearly with wind speed. Thus, a rated wind speed of 4.5 m/s corresponds to a rotor speed of 230 rpm. The flutter speed of 6 m/s (total air speed 34 m/s) corresponds to a rotor speed of 315 rpm. In this section, operating conditions will be designated by the operating rotor speed.

### 6.1   System identification and stabilising controller

Initially, the response of the wind turbine blade to flap actuation is studied and compared with the simulations. Open-loop

identification experiments are conducted in the pre-flutter regime, 200-300 rpm, with a zero-mean white noise (maximally +/- 500 V) imposed on the piezobenders. Predictor-based subspace identification (PBSID) Van der Veen et al. (2013) is performed using the acquired data to obtain the transfer function between the tab actuation and the flap angle and blade root load measurements. The transfer functions are depicted in Figures 17 and 18.

It can be observed that significant phase loss occurs in the transfer from the actuator to the blade root loads. This implies

that stabilising the system using the measurements from the root loads poses a control challenge, and it may prove difficult in the case of uncertain systems to guarantee robust stability in the unstable post-flutter region. Further, it also motivates the use of local load sensors to enhance load attenuation capabilities. On the other hand, the phase loss in the transfer between the actuator and the flap angle measurement is minimal. This collocated sensor is hence ideal for system stabilisation in the post-flutter region. A simple classically tuned controller is used for stabilisation, it is not designed for load reduction, and is

hence not optimal. It is described in continuous time as follows:

$$K = \underbrace{0.0001}_{\text{Static gain}} \underbrace{\frac{s/0.001+1}{s/10+1}}_{\text{High pass}} \underbrace{\frac{s^2+0.001s*50/2\pi+(50/2\pi)^2}{s^2+0.1s*50/2\pi+(50/2\pi)^2}}_{\text{notch for 50 Hz electrical back-coupling artefact}} \underbrace{\frac{1}{2\pi s/100+1}}_{\text{Low pass}}.$$ (19)

This controller is now used in closed-loop for studying system behaviour beyond flutter. Closed-loop identification experiments are performed in a similar manner and the transfer functions are obtained using PBSID, also shown in the Figures 17 and 18. In all identification experiments, the variance accounted for (VAF) Van der Veen et al. (2013) values exceed 60%. The

dynamic behaviour can be seen to follow the predicted aeroelastic response from Figures 13 and 14. The frequency of the blade flexible mode remains more or less constant, Figure 17, however the damping goes on reducing until it is unstable at 340 rpm, as indicated by the sharp peak at 74 rad/s (11.8 Hz). On the other hand, Figure 18 shows that the frequency of the rigid-body mode increases with wind speed, along with the damping, as predicted in NASTRAN. Finally, system identification shows that the control authority of the flaps is low at low frequencies, but increases substantially at and beyond 2P (8Hz), making it

suitable for reducing 2P loads and loads induced by turbulence.




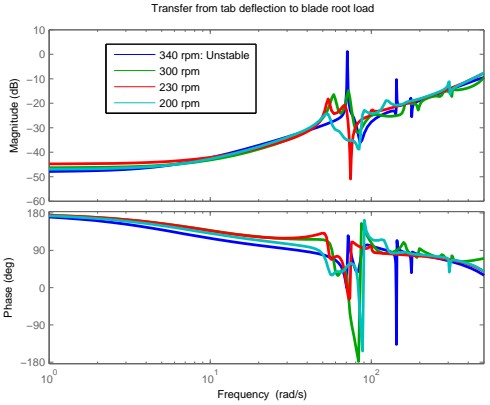
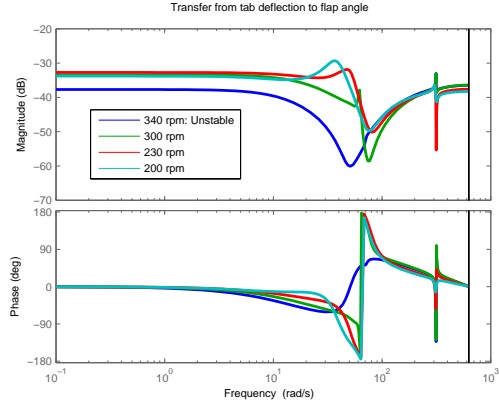

**Figure 17.** Transfer from piezobender actuators to blade root loads, different wind speeds.

**Figure 18.** Transfer from piezobender actuators to free floating flap angle, different wind speeds.

## 6.2 Optimal IFT tuning for constant wind speeds: Pre-flutter

The next step was to study the effect of the IFT load controllers for combined pitch and flap control. The block diagram for the load controllers is shown in Figure 16. Accordingly, the pitch and flap actuation signals were combinations of 1P and 2P sinusoidal basis functions, respectively. The basis functions are scheduled on the azimuth and are hence phase-locked. IFT was

used to train the amplitudes of these basis functions; thus, with two basis functions for each frequency and each blade, for pitch and flap control both, a total of eight gains were required to be tuned.

The IFT process was first studied for a constant operational speed. Selected results, at an operational speed of 230 rpm, are presented here, although similar results were also observed throughout the operational range. The convergence of the controller gains and the IFT cost criterion can be seen in Figures 19 and 20. It can be seen that, within ten minutes, the controller gains

converge to their optimal values. The performance of the controller after convergence can be visualised in Figures 21 and 22. The figures show that the actuation demanded, both pitch and flap, is purely sinusoidal, as constrained by the respective basis functions. Further, the load components in the blade load spectrum at the frequencies 1P and 2P are almost entirely eliminated by the pitch and flap action respectively. Thus, IFT is successful in tuning the controllers as required.

One final point of note is that the converged gains for the two blades are not exactly antisymmetric, this is especially

pronounced for the flap actuation. The primary reason for this is a difference in the manufacture of the two blades. Specifically for the flap dynamics, for the scaled blade, a difference in the order of a few grammes in its weight distribution can strongly alter system dynamics and even prepone the onset of flutter.

Commercially manufactured blades are ideally expected to be identical; they would require antisymmetric pitch action and identical flap action for load attenuation, as produced by a conventional IPC controller Bossanyi (2003). Such a controller

does not achieve optimal load reduction in the case there are discrepancies in blade manufacture or aging. The IFT controller




designed above is thus shown capable of accounting for blade asymmetry and adjusting control action for maximising load reduction.

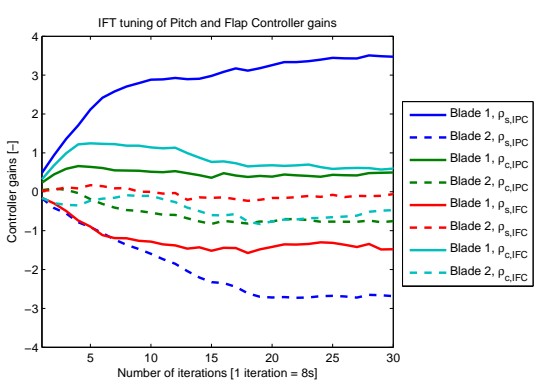

**Figure 19.** Convergence of controller gains over iterations

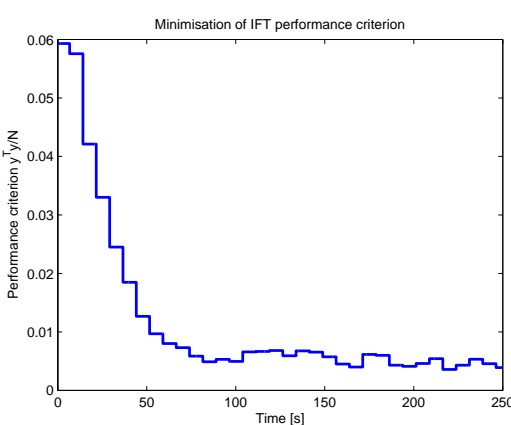

**Figure 20.** Minimisation of IFT cost criterion over iterations

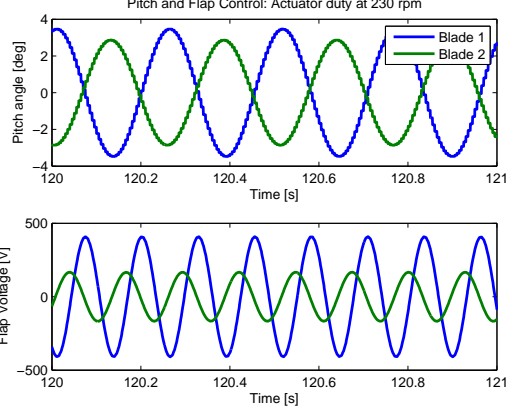

**Figure 21.** Actuator duty cycles of optimised controller
(PRE-FLUTTER)

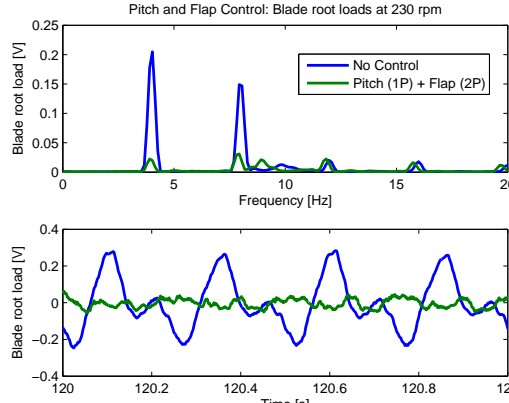

**Figure 22.** Load reductions achieved by optimised controller
(PRE-FLUTTER)

### 6.3 Optimal IFT tuning for constant wind speeds: Post-flutter

Next, the free-floating flap is connected in series with the stabilising PID controller described above and the wind turbine
5 is run at an operational speed of 330 rpm, in the post-flutter region. Since it is not optimally tuned, this controller does not maximise load reductions. Hence, IFT is used to tune the feedforward load reducing pitch and flap controller gains in a manner similar to the previous experiments; however in this case the underlying plant is the stabilised post-flutter wind turbine in closed-loop with the PID controller. From Figures 23 and 24 it can be seen that the optimised IFT controller gains are still



able to achieve load reduction even in this highly challenging unstable operational regime. The Figure 23 shows that the pitch controller no longer issues antisymmetric commands; a traditional IPC controller is no longer adequate in this region. Further, the flap command has already reached its maximum limits of +/- 500 V.

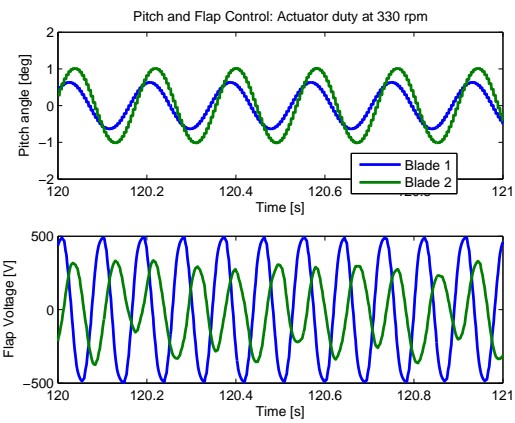

**Figure 23.** Actuator duty cycles of optimised controller (POST-FLUTTER)

**Figure 24.** Load reductions achieved by optimised controller (POST-FLUTTER)

### 6.4 Optimal IFT gain-schedule tuning for varying wind speeds

For an in-field wind turbine, controller gain optimisation cannot be implemented considering the wind speed to be constant. Hence, the gain scheduling approach described in the previous section is followed, where, instead of the absolute values of the controller gains, the coefficients of a gain schedule, $\rho^{[0]}$ and $\rho^{[1]}$ are optimised based on the IFT experiments. This method is tested in the wind tunnel, with a varying operational speed profile as depicted in Figure 25. The convergence of the gain schedule coefficients can be seen in Figures 26 and 27. It can be seen that in the first 100 seconds, since the wind speed is

constant, a good gain schedule cannot be identified owing to a lack of persistency of excitation in the scheduling parameter. However, as the wind speed changes beyond this point in time, the gain schedule rapidly converges to an optimum. The gain schedule finally achieved is compared with the optimal controller gains obtained from the previous set of experiments in Figures 28 and 29. It can be seen that for the pitch controller, the linear gain schedule obtained is a good fit to the optimal values obtained at constant wind speed. On the other hand, the flap controller optimal gains show a non-linear variation with

wind speed and the linear gain schedule obtained achieves reduced goodness of fit.

Herewith, the combined pitch and flap controller has been shown to be able to reduce blade loads both in pre- and post-flutter conditions. Further, an optimal gain schedule for these controllers is automatically tuned online using IFT in varying wind conditions.




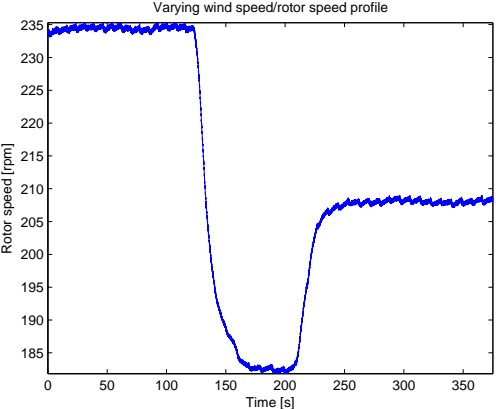

**Figure 25.** Varying operational speed for optimisation of
gain schedule.

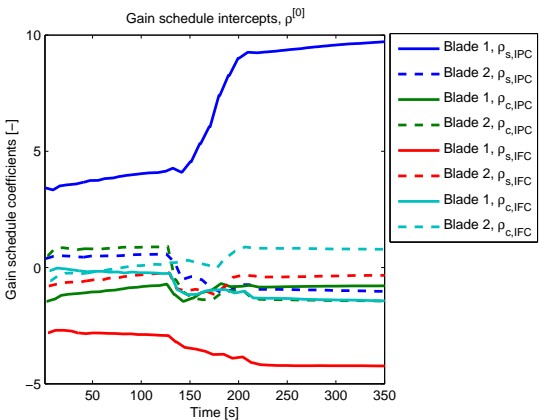

**Figure 26.** Optimisation of gain schedule intercepts for vary-
ing wind speed conditions.

**Figure 27.** Optimisation of gain schedule slopes for varying
wind speed conditions.

## 7 Conclusions

A successful experimental proof of concept has hereby been achieved for the first time of free-floating flaps for wind turbines
and of combined pitch and flap control for blade load mitigation.

Free-floating flaps were designed for the first time for the application of wind turbine load control. Numerical aeroelastic
5 analysis concluded that such flaps show significant control authority in the desired frequency band (2P and beyond). However
the additional degree of freedom couples aerodynamically with the flapwise flexible mode of the blade and causes flutter at
low wind speeds, just outside the design envelope. Using a feedback controller, the blade can be stabilised in the post-flutter
region. Both of these results were validated experimentally in the wind tunnel.





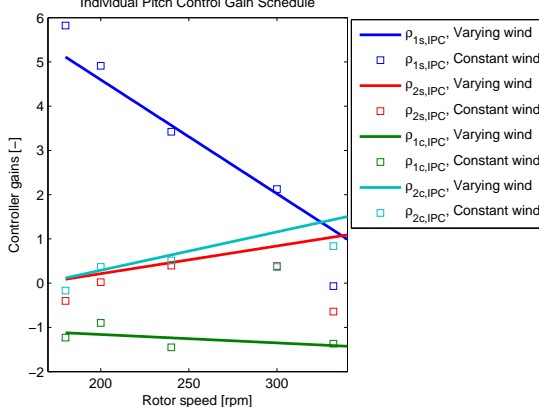
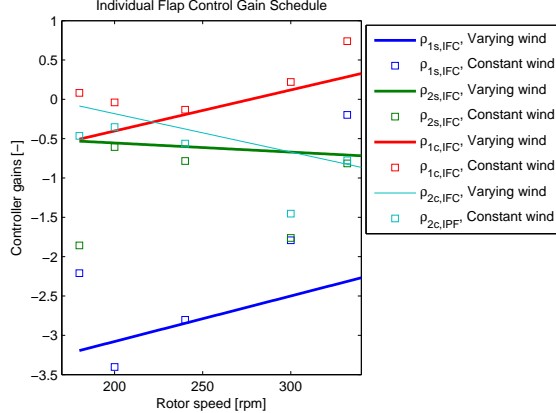

**Figure 28.** Gain schedule at varying wind speeds versus optimal gains at constant wind speed: Pitch Control.

**Figure 29.** Gain schedule at varying wind speeds versus optimal gains at constant wind speed: Flap Control

Blades were manufactured using the novel combination of 3d-printing with carbon fibre layup, and instrumented with free-floating flaps close to the blade tip. The concept of Iterative Feedforward Tuning of the gains of phase-locked basis functions was used to achieve blade load reductions. The pitch control action was composed of a superposition of 1P sinusoidal basis functions, while for the flap control action, 2P sinusoidal basis functions were used. It was shown that, for a constant pre-flutter

wind speed, ideal rejection of 1P and 2P loads in the blade load spectrum could be achieved with combined pitch and flap control. Further, at a post-flutter wind speed, the system was connected in closed-loop with a stabilising PID controller using collocated feedback. Iterative Feedforward Tuning was able to optimise, in this unstable regime, the load control gains of the pitch and flap control action for this closed-loop plant. Load rejection was also achieved in the challenging post-flutter regime, although the flap actuation duty reached close to its physical limits under these conditions. Finally, for the case of varying wind

speed conditions, the IFT methodology was able to autonomously synthesise an optimal linear gain schedule, in real time, for the combined pitch and flap controller. Such a gain schedule was found to be near-optimal for a large portion of the range of operation.

With the inclusion of flap control, the individual pitch controller can focus purely on 1P load attenuation; this reduces pitch activity, especially in the high-frequency region of the spectrum, and can enhance the longevity of the pitch actuation

mechanism.

With free-floating flap control with variable pitch wind turbines hereby validated, future work will primarily concern the LPV (linear parameter-varying) modelling and validation of the augmented turbine blade, and the synthesis of an optimal LPV controller. Also, turbulent and gust load mitigation with local flap control would form the next step in the study of wind turbines with free-floating flaps.



*Acknowledgements.* This work was supported by the INNWIND.EU Project, an EU Consortium with Academic and Industrial Partnership for Innovations in Wind Energy.

We would like to thank ir. Kees Slinkman and ing. Will van Geest (Delft Centre for Systems and Control) for their help in the design and assembly of the experimental setup.



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
