# Peer review of "Wind tunnel tests with combined pitch and free-floating flap control: Data-driven iterative feedforward controller tuning"

_Wind Energy Science, 2016_

## Referee Comment (RC1) · Anonymous Referee #1 · 10 Jul 2016

General comments

The paper presents, in the first part, a wind tunnel setup equipped with a combined pitch and free-floating flap control. In the following part of the paper the authors present an iterative feedforward controller tuning used to optimize the control law online an automated manner in order to maximize the load reduction. The topic is very important for the wind energy community and the tools/techniques represent the state of the art, so that the publication on a journal is recommended, even if some modifications are required.

Specific comments

[Figure]

+ Page 1, line 7. "The inertia of the flaps was tuned ...". Nowhere in the paper the authors present this inertia tuning. Pleas comments this. + Section 2. The authors write they have scaled the INNWIND.EU rotor, but the scaling laws are not reported (time ratio, length ratio, Lock, Reynolds, etc.). Moreover, the scaled model is a 2-bladed rotor, the INNWIND.EU a 3-bladed. The only information provided is (page 3, line 24-25) is on the first blade freq. wrt the 1P (>3.5) which is a standard value for all 3-bladed rotor (to avoid intersections in the Campbell diagram between the blades and the 1/2/3P). It looks like the scaled model is a proof of concept of technology more than a scaled model of a full scale wind turbine, so that the table 1 and the reference to the INNWIND rotor should be removed. Otherwise the authors must give more info about the scaling. + Section 2.1. No information is provided about the aerodynamic design: how the authors have chosen airfoils, chord and twist distribution? May the authors include some more info about this and some more data about the overall performances (power coefficients VS TSR, for instance). + Page 5, lines 5-10. The authors present a mismatch between the measured and calculated structural behavior. The reason of this has been identified in the anisotropic behavior of the real blade not modeled in the isotropic FEM model. The authors should comment why they have not tried to identified this anisotropic behavior on some specimens (as done for Figure 1...) and then used an anisotropic FEM model. + Page 7, line 10. "...adds an additional rigid-body degree of freedom": this comments is unnecessary because this is well known; moreover this comment does not need to be supported by two (auto)citations. + Page 7, lines 18-22. The discrepancy (about 20%) between the two mathematical models is an error in the modeling: a correct FEM model and a correct cross-sectional code + beam model can give the same (correct) results. In a journal paper this should be correct. Moreover, both the models return huge error wrt the real one (see previous point...). + Page 9, lines 1-2. This is not clear. The airspeed 36m/s refers to the scaled or the full scale model (it looks the full one...)? 340rpm is the scaled one. Probably the authors should present a regulation trajectory of the (scaled) wind turbine (i.e. rotor speed VS wind) + Page 9, figures 13-14. The flutter analysis presented here looks more the one used

for fixed wing (i.e. uniform airflow on the blade, constant AoA, no rotation). Is it also applicable on a rotation blade? Please add some comments in the paper about this flutter analysis. The wind speed on the x-axis refers to the scaled model? + Page 16, lines 4-7. Again more information about the operation of the model is necessary: if the rated speed is 4.5m/s at 230rpm, at 6m/s the rotor speed of a classical variable-speed pitch-regulated wind turbine is again 230rpm (i.e. in the above-rated region the rotor speed is kept constant). The authors must better define the regulation of the model.

Technical corrections/comments + Some figures may be more readable if different line styles are used (i.e. solid, dash-dotted, dotted, etc..). This helps if read on black/white copies or by color-blind person…. + Page 6, line 19. Add extra space: "Finally,an" + Page 8, fig 9-10. Please check these figures, because they look inverted (Fig. 9 looks the first edgewise mode…) + Page 12, line 7, correct "currrent" + Sections 6.2, 6.3, 6.4. In the titles the word "tuning" should be removed since already included in the acronym "IFT" + Figures 21-24: why the words PRE-POST FLUTTER are uppercase?

---

## Referee Comment (RC2) · Anonymous Referee #2 · 18 Jul 2016

Author proposed a control strategy to alleviated wind turbine load. This control strategiy is based on the combined used of full-span pitch and trailing-edge flap control. This strategy is asses on real experiments that tuned to be instable just beyond the operational envelope of the wind turbine. This article begins with a brief state of art on the problematic. Author deals with all benefits of the different control strategy it will be used, the control objectives that could be reached (reduction load upto 70%). In the end of this part the author sum-up the three main contribution of this article: - Design and manufacturing of the wind turbine blades, - Demonstration of the potential of the combination of free-floating flaps and individual pitch control in terms of load alleviation, - A novel iterative feedback tuning algorithm. Please precise what is new,

the combination of the 2 control strategy? The real time aspect? Referring to your personal bibliography some of these aspects has already been treated. The reader should clearly be able to locate this new article in your scope. Comments to this part: - Line 3 page 2: please defined 1P in this chapter, the definition of this acronym comes in chapter 2 (too late), - You affirmed that "Data-driven controller may be able to achieve greater optimality of performance without the excessive conservatism of the true robust design"; It seems you want to compare totally opposite methodology. Data-driven has no proof of robustness and/or optimality and the performance of the controller depend of the number of data-acquisition set you used to determine your controller. This methodology has advantages of deleted identification part and to be simple to use and implemented but does your non-linear system is not a linear piecewise system? Does you have some robust control methodology that you could use? - You must précised clearly what is new compared to your bibliography. The combination of the methodology? The real time aspect? If we read some paper of your bibliography it seems that all aspect is already treated in your previous article. After the brief state of art on the problematic (wind turbine load, data-driven control…) the author proposed a chapter on the blade design and manufacturing. Comments to this part: - This chapter must be design to offer real investigation and comprehension of the design and manufacturing of blades for people that cannot be accustomed to these topics. Why this parts is important to understand the following parts of this article? - Figure 3 is cited befor figure 1 and 2, - Line 8 page 4 : what does you mean with this word "qua"? - Line 24 page 4 :what does you mean with CAD? - Figure 2 : can you explain where is the sample and what is around the sample? - You found that the blade designed using CAD software is 30% stiffer than the actual manufactured blade but doesn't explain why. Please avoided postulated affirmation and concretely explained the difference and the consequence on the announced gain of 70%. 30 % more stiffness of the blades does not seem to be reflected in the same way on the resonance frequency that not only down 13%. Can you explain why? Chapter 3 and 4 deals with the aeroelastic blades analysis and the wind turbine test bench description. These chapters are well written and

usefull to the comprehension. Chapter 5 is concerned by the control algorithm. This chapter should be rewritten, taking care to present things in the clearest way possible: - Line 14 page 11 you talk about an optimal control action, where is the proof of the optimality of the control action? Do you have references that assess that the control action is optimal? It's an global optimality or an operating point optimal controller? - Line 19 page 11, Does the plant or the controller is assumed be still LPV? - Line 20 page 11: What is the consequence of the assumption of constant wind speed during each set of IFT experiments? - Line 32 page 11 you wrote that Ak, Bk, Ck, Dk are considered unknown so how you assess that this same matrices could be written as equation 3. What does the bracket [0] or [1] means? - Equation 5 page 12, you use qk and write line 22 page 12 that it will be described in the next section but in section 5.2 you wrote that qk is equal to zero. What the interest of this variable? This variable doesn't appear on the block diagram figure 16. - Where is the global control scheme (IPC+IFC+SYSTEMS+PID...)? - Line 9 page 14: $\mu$ is not defined (a residual of Sachin T. Navalkar et al. / IFAC-PapersOnLine 48-26 (2015)?) - 1st paragraph Page 15 : you precise that your controller is optimal for the operating point that assess you can't compared it to robust control. What happen when the wind speed is between two operating point you use to find your controller? Chapter 6 deals with results. Results are good.
* * *

---

## Author Comment (AC1) · 8 Aug 2016

[12pt,logo]tubrief  bibunits

Response to reviewers

Delft Center for Systems and Control

Address
Mekelweg 2 (3ME building)
2628 CD Delft
The Netherlands

www.dcsc.tudelft.nl Sachin T. Navalkar

To
Reviewer 1
*Wind Energy Science*

Dear Reviewer,

First of all, the authors would like to thank the reviewer for their positive and constructive feedback. We believe that the comments have helped us improve the quality of the paper. In our attempt to account for the comments, we plan to revise different aspects of the paper. The objective of this document is to respond to the points raised by the reviewer and to provide a detailed overview of the changes being made to the paper.

Yours sincerely,

Response to comments of Reviewer 1

**1 Response to comments of Reviewer 1**

1. Page 1, line 7. "The inertia of the flaps was tuned..." Nowhere in the paper the authors present this inertia tuning. Please comment on this.
During the design of the experiment, the flutter analysis conducted on the numerical model of the blade showed high sensitivity to the exact value of the flap inertia about its free hinge axis. As such, a sensitivity analysis was conducted to ensure that the blade does not enter the flutter regime prematurely, and that the system is stable in the intended regime of operation. This sensitivity analysis was omitted from the paper, but, as per the feedback of the reviewer, could be included in the revised version of the manuscript.

2. Section 2. The authors write they have scaled the INNWIND.EU rotor, but the scaling laws are not reported (time ratio, length ratio, Lock, Reynolds, etc.). Moreover, the scaled model is a 2-bladed rotor, the INNWIND.EU a 3-bladed. The only information provided is (page 3, line 24-25) is on the first blade freq. wrt the 1P (>3.5) which is a standard value for all 3-bladed rotor (to avoid intersections in the Campbell diagram between the blades and the 1/2/3P). It looks like the scaled model is a proof of concept of technology more than a scaled model of a full scale wind turbine, so that the table 1 and the reference to the INNWIND rotor should be removed. Otherwise the authors must give more info about the scaling.
The authors agree with the reviewer that the description of the scaling, especially with relation to the INNWIND.EU rotor, is inadequate. However, due to practical considerations, scaling was only done to ensure that the frequency ratio of the first blade mode with respect to 1P remained constant. As such, the authors agree that this scaled turbine simply represents a model for the proof of concept, and the reference to the INNWIND.EU rotor will be removed.

3. Section 2.1. No information is provided about the aerodynamic design: how the authors have chosen airfoils, chord and twist distribution. May the authors include

some more info about this and some more data about the overall performances (power coefficients VS TSR, for instance)?

The reviewer raises an important point, it needs to be mentioned that the aerodynamic design of this specific rotor follows the design approach of Van Wingerden et al. (2011), where a similar rotor, but with conventional flaps (and no pitch control) was tested under similar experimental conditions. The authors of the current paper will provide the necessary references, for instance "Design of a scaled wind turbine with a smart rotor for dynamic load control experiments" by Hulskamp et al, 2011, and related, where a detailed analysis of the aerodynamic design and other performance characteristics can be found.

4. Page 5, lines 5-10. The authors present a mismatch between the measured and calculated structural behavior. The reason of this has been identified in the anisotropic behavior of the real blade not modeled in the isotropic FEM model. The authors should comment why they have not tried to identified this anisotropic behavior on some specimens (as done for Figure 1...) and then used an anisotropic FEM model.

As noted, there is indeed a difference between the measured and calculated structural behaviour. The manufacturer-published values of anisotropic material properties were used to estimate the stiffness of the blade. However, it was observed that the stiffness of the blade is not significantly affected by using an anisotropic model. The authors retested the stiffness of the blade, and it was found that the clamp used for fixing the blade root was not ideal, but allowed rigid-body rotation of the blade. The rotation of the blade root with increasing load was measured, and compensated for in order to correctly estimate the actual stiffness of the blade. [IF THE FIGURE IS NOT VISIBLE, PLEASE REFER TO THE SUPPLEMENTARY PDF ATTACHED]. The corrected figure will be used in the revised manuscript.

5. Page 7, line 10. "..adds an additional rigid-body degree of freedom": this com-

stiffplotmod.png

ment is unnecessary because this is well-known; moreover this comment does not need to be supported by two (auto)citations.

The authors agree that the statement is self-evident, and will be removed.

6. Page 7, lines 18-22. The discrepancy (about 20%) between the two mathematical models is an error in the modeling: a correct FEM model and a correct cross-sectional code + beam model can give the same (correct) results. In a journal paper this should be correct. Moreover, both the models return huge error wrt the real one (see previous point...).

Indeed, there is a large difference in the modal description of the two mathematical models. This was a result of a difference in the clamping conditions: while the blade root section was completely clamped in the NASTRAN model, only the connecting nut was clamped in the Solidworks model. When this discrepancy was removed by correctly clamping the Solidworks model, the difference between the two models reduces to a large extent; the recalculated difference being less than 2.5%.

- 1st Flapwise frequency: 18.97 Hz (Solidworks), 19.44 Hz (NASTRAN)
- 1st Edgewise frequency: 78.37 Hz (Solidworks), 76.67 Hz (NASTRAN)
- 2nd Flapwise frequency: 84.8 Hz (Solidworks), 87.88 Hz (NASTRAN)

The revised manuscript will contain this updated modal analysis. As far as the difference between the numerical and experimental model is concerned, it should be noted that the numerically calculated values are blade frequencies; the rotor modal frequencies measured in practice are also influenced by the blade connection flexibility, motor stiffness and hub flexibility, and as such are necessarily lower than the numerically calculated blade modal frequencies. The difference, or uncertainty, in rotor modal frequency estimation is a powerful motivation for the choice of a self-tuning regulator as the load controller, since it can adapt to the true system parameters automatically. This explanation will be included in the manuscript.

7. Page 9, lines 1-2. This is not clear. The airspeed 36m/s refers to the scaled or the full scale model (it looks the full one...)? 340rpm is the scaled one. Probably the authors should present a regulation trajectory of the (scaled) wind turbine (i.e. rotor speed VS wind).
The terminology "air speed" does appear unclear, it is the resultant air speed incident on the blade, which is a combination of the inflow wind speed and the wind speed induced by the rotation of the blade. This definition will be made clear in the revised manuscript. The regulation trajectory of the scaled wind turbine is linear, and as such can be described by a single coefficient denoting its slope,

which is 51.1 rpm/(m/s). This value will also be stated clearly in the revised manuscript.

8. Page 9, figures 13-14. The flutter analysis presented here looks more the one used for fixed wing (i.e. uniform airflow on the blade, constant AoA, no rotation). Is it also applicable on a rotation blade? Please add some comments in the paper about this flutter analysis. The wind speed on the x-axis refers to the scaled model?

As pointed out by the reviewer, the flutter analysis presented is indeed carried out for a fixed wing, with uniform airflow and constant angle of attack. The effect of rotation is only included in the sense that the incident wind speed is not the inflow wind speed, but rather, the total air speed as defined in the previous comment. This forms a first approximation to the true flutter behaviour of the rotating blade, the inflow conditions along the span of the blade are held constant to those obtained at the tip section. While the error in the aerodynamic forces increases for the sections radially inboard, these sections also undergo smaller structural motions and hence contribute to a progressively smaller extent to the aeroelastic behaviour and hence the modal analysis of the blade as a whole. Further, the blade is typically twisted such that the angle of attack remains constant throughout the blade span; this is approximated by using aerodynamic panels with no twist, and with a constant angle of attack, along the span of the blade. As such, it is assumed that the flutter analysis of the non-rotating blade is a good approximation of the dynamic behaviour of the actual experimental blade. This explanation will be included in the revised manuscript. Finally, it should be pointed out that the "air speed" in the Figures 13 and 14 refers to the total air speed, as defined in the previous comment; this will also be clarified.

9. Page 16, lines 4-7. Again more information about the operation of the model is necessary: if the rated speed is 4.5m/s at 230rpm, at 6m/s the rotor speed of a classical variable-speed pitch-regulated wind turbine is again 230rpm (i.e. in the

above-rated region the rotor speed is kept constant). The authors must better define the regulation of the model.

Thank you for this remark. As described in the second paragraph of Section 4, the load resistance connected to the wind turbine generator is kept constant; this constant load operation implies that the rotor speed increasely linearly with the wind speed at a rate of 51.1 rpm/(m/s). This is indeed not similar to the operation of a classical variable-speed pitch regulated wind turbine, where collective pitch control is used to ensure that the rotor speed remains constant (above-rated) irrespective of the change in wind speed. However, the constant resistance operation of the turbine serves three main purposes: it describes the effect of (temporary) overspeeds which may cause the turbine to enter flutter, it emulates the behaviour of the turbine under below-rated conditions, and it describes the potential of an adaptive control strategy that may be required to retune itself under varying operating conditions. This description and motivation of constant load operation will be clarified in the revised manuscript.

10. Some figures may be more readable if different line styles are used (i.e. solid, dash-dotted, dotted, etc..). This helps if read on black/white copies or by color-blind person. Page 6, line 19. Add extra space: "Finally,an".
    The figures and the typo will be revised as advised.

11. Page 8, fig 9-10. Please check these figures, because they look inverted (Fig. 9 looks the first edgewise mode...).
    The figures were checked and found to be correct. The flapwise modes display deformation perpendicular to the aerodynamic panels, while the lead-lag mode displays deformation in the plane of the aerodynamic panels. This description will be updated in the revised manuscript.

12. Page 12, line 7, correct "currrent". Sections 6.2, 6.3, 6.4. In the titles the word "tuning" should be removed since already included in the acronym "IFT". Figures

21-24: why the words PRE-POST FLUTTER are uppercase?
The corrections will be made as advised by the reviewer, we thank them once again for their constructive feedback.

---

## Author Comment (AC2) · 8 Aug 2016

Response to reviewer 2

Delft Center for Systems and Control

Address
Mekelweg 2 (3ME building)
2628 CD Delft
The Netherlands

www.dcsc.tudelft.nl Sachin T. Navalkar

To
The Anonymous Reviewer 2
*Wind Energy Science*

Dear Reviewer,

First of all, the authors would like to thank the reviewer for their positive and constructive feedback. We believe that the comments have helped us improve the quality of the paper. In our attempt to account for the comments, we plan to revise different aspects of the paper. The objective of this document is to respond to the points raised by the reviewer and to provide a detailed overview of the changes being made to the paper.

Yours sincerely,

Response to comments of Reviewer 2

[Figure]

**1 Response to comments of Reviewer 2**

1. Please precise what is new, the combination of the 2 control strategy? The real time aspect? Referring to your personal bibliography some of these aspects has already been treated. The reader should clearly be able to locate this new article in your scope.
   Thank you for this remark, the original introduction was indeed unclear regarding the precise aspect in which the paper represents an advancement of the state of the art. To answer the question, three main novelties can be found in this paper:

   (a) This is the first experimental demonstration of combined pitch and flap control.

   (b) This is the first experimental demonstration of free-floating flaps applied to rotating wind turbine blades. This is also the first time their potential has been demonstrated for load reduction in wind turbines experimentally. Further, this is the first time that free-floating flaps have been shown to induce flutter on wind turbine blades experimentally.

   (c) This is the first experiment where IFT has been devised and implemented for adaptively tuning the gain schedule of a nonlinear (LPV) system.

   Since these novelties are not clear, they will be mentioned explicitly in the introduction of the revised manuscript.

2. Line 3 page 2: please defined 1P in this chapter, the definition of this acronym comes in chapter 2 (too late),
   The term 1P will be defined as the rotor speed in the introduction of the revised version of the manuscript.

3. You affirmed that "Data-driven controller may be able to achieve greater optimality of performance without the excessive conservatism of the true robust design";

It seems you want to compare totally opposite methodology. Data-driven has no proof of robustness and/or optimality and the performance of the controller depend of the number of data-acquisition set you used to determine your controller. This methodology has advantages of deleted identification part and to be simple to use and implemented but does your non-linear system is not a linear piecewise system? Does you have some robust control methodology that you could use?

The authors agree completely with the reviewer that the data-driven strategy described in this paper has no proof of robustness or optimality, and depends strongly on the realisations of the data and wind speed during operation. In case one were to approach the controller design from a traditional robust mindset, one would be required to formulate an accurate linear parameter-varying or piecewise linear description of the system, which would undoubtedly be subject to significant parametric and dynamic uncertainties. Formulating a robust controller based on such an uncertain plant description could yield a strongly conservative controller that is perhaps not able to achieve the maximum possible load reductions. The advantage of the IFT approach is that, besides being simple, it uses input-output data to tune itself to optimise a simple user-defined criterion. Since a feedforward approach is used, the IFT controller cannot destabilise the plant; in the best case, it could perhaps achieve load reductions that a conservative robust controller may not be able to reach.

This discussion will be repeated in the revised manuscript to clearly compare the proposed control strategy with the robust control approach.

4. You must précised clearly what is new compared to your bibliography. The combination of the methodology? The real time aspect? If we read some paper of your bibliography it seems that all aspect is already treated in your previous article. Please refer to the discussion related to the first comment.

5. This chapter must be design to offer real investigation and comprehension of the design and manufacturing of blades for people that cannot be accustomed to

these topics. Why this parts is important to understand the following parts of this article?

The authors apologise for the lack of clarity in describing the aim of Section 2. In principle, this section describes the design of the experimental setup, and provides details regarding the materials, method of manufacture and assembly. The authors believe that this is important, since it is the first time that a wind turbine blade has been manufactured to incorporate free-floating flaps. Primarily, the destabilising effect of the free-floating flap is studied in detail, and the parameters are tuned such that the blade is close to, but not beyond, the flutter point in order that maximal control authority is achieved. This reasoning will be included in the introductory paragraph of Section 2.

6. Figure 3 is cited befor figure 1 and 2.
   This will be rectified by reordering the figures.

7. Line 8 page 4 : what does you mean with this word "qua"?
   "Qua" has the meaning "in terms of".

8. Line 24 page 4 :what does you mean with CAD?
   CAD stands for Computer-Aided Design, the expansion of the acronym will be included in the revised manuscript.

9. Figure 2 : can you explain where is the sample and what is around the sample?
   The top grey rectangle and the two black rectangles below it are the 3d-printed plastic samples, bonded with carbon fibre. They are placed on a sandstone-coloured desktop, this forms the background. This will be made clear in the caption.

10. You found that the blade designed using CAD software is 30% stiffer than the actual manufactured blade but doesn't explain why. Please avoided postulated affirmation and concretely explained the difference and the consequence on the

announced gain of 70%. 30 % more stiffness of the blades does not seem to be reflected in the same way on the resonance frequency that not only down 13%. Can you explain why?

The authors retested the stiffness of the blade, and it was found that the clamp used for fixing the blade root was not ideal, but allowed rigid-body rotation of the blade. The rotation of the blade root with increasing load was measured, and compensated for in order to correctly estimate the actual stiffness of the blade. [IF THE FIGURE IS NOT VISIBLE, PLEASE REFER TO THE ATTACHED SUPPLEMENTARY PDF]. The corrected figure will be used in the revised manuscript.

```
stiffplotmod.png
```

The announced gain of 70% is an experimentally measured quantity, and has no relation to the numerical models (that do not predict load-reduction capabilities). As far as the difference between the numerical and experimental model is concerned, it should be noted that the numerically calculated values are blade frequencies; the rotor modal frequencies measured in practice are also influenced by the blade connection flexibility, motor stiffness and hub flexibility, and as such are necessarily lower than the numerically calculated blade modal frequencies. These issues will be clarified in the revised manuscript.

11. Line 14 page 11 you talk about an optimal control action, where is the proof of the optimality of the control action? Do you have references that assess that the control action is optimal? It's an global optimality or an operating point optimal controller?
The reviewer points out an important issue: IFT has no theoretical proof of global optimality. Indeed, if IFT is used for tuning a feedback controller, there is no guarantee that the closed-loop system will be stable. Since the paper discusses a feedforward controller, this is not an issue of concern. Further, if the step-size in the gradient descent algorithms is too large, the parameter tuning process may become unstable. These issues have been dealt with by Hjalmarsson in the reference "Iterative feedback tuning: an overview." This paper will be referenced in Section 5 in the revised manuscript. The "optimality" described in this section refers specifically to the optimisation in a local sense, of the user-defined cost function. This will also be made explicitly clear in the revised text.

12. Line 19 page 11, Does the plant or the controller is assumed be still LPV?
The original statement is ambiguous. The plant, being the wind turbine, is at all times LPV. For the case where the wind conditions are held constant in the wind tunnel, an LTI controller is tuned by IFT for that specific operating point of the LPV plant. On the other hand, for the case where the wind conditions are allowed to vary in the wind tunnel, a gain-scheduled controller is tuned by the IFT algorithm developed in this paper. This schema will be explicitly mentioned in the introduction of Section 5.
13. Line 20 page 11: What is the consequence of the assumption of constant wind speed during each set of IFT experiments?

The assumption of constant wind speed implies that ordinary IFT can yield an optimal load-reducing controller only for that specific operating point. Such a controller may not achieve the highest possible load reductions, or may even increase loads, at other operating points. It is for this reason that the ordinary IFT process has to be repeated for different constant wind speeds, or an IFT gain schedule has to be generated for a varying wind speed. This will be made clear in the revised manuscript.

14. Line 32 page 11 you wrote that Ak, Bk, Ck, Dk are considered unknown so how you assess that this same matrices could be written as equation 3. What does the bracket [0] or [1] means?

$A_k$, $B_k$, $C_k$ and $D_k$ are all considered unknown but assumed to admit a specific LPV structure, often used for modelling wind turbines, as in the PhD thesis of Van Wingerden; a reference to this thesis will be provided in the revised text. The terms $A^{[0]}$ and $A^{[1]}$ signify the unknown components of $A_k$, one which is constant over time, and one that varies linearly with the wind speed. As per Van Wingerden 2008, there should be one more term that varies with $V_k^2$, however the influence of this term is small and it is neglected in this paper. This description will be added to the revised text.

15. Equation 5 page 12, you use qk and write line 22 page 12 that it will be described in the next section but in section 5.2 you wrote that qk is equal to zero. What the interest of this variable? This variable doesn't appear on the block diagram figure 16.

The authors thank the reviewer for pointing out this issue. While $q_k$ is indeed zero for the reference experiment, it is non-zero for the gradient experiment of IFT, please see equation (11). As such, it is important for determining the gradient of the performance criterion with respect to the controller parameters. The authors

will update Figure 16 such that this signal is included in the block diagram.

16. Where is the global control scheme (IPC+IFC+SYSTEMS+PID...)?
A new block diagram will be included in the revised manuscript that extends the block diagram scheme of Figure 16 with the stabilising collocated PID control.

17. Line 9 page 14: $\mu$ is not defined (a residual of Sachin T. Navalkar et al. / IFAC-PapersOnLine 48-26 (2015)?)
The authors thank the reviewer for pointing out this typo, $\mu_*$ should indeed be replaced by $V_*$.

18. 1st paragraph Page 15 : you precise that your controller is optimal for the operating point that assess you can't compared it to robust control. What happen when the wind speed is between two operating point you use to find your controller?
The reviewer points out a shortcoming of the two control approaches used in the paper:

   • For the case where LTI controllers are devised for constant operating points, the control action for an intermediate speed is obtained by interpolating between the gains of the closest wind speeds. This is the approach followed by most industrial gain-scheduled controllers. For a highly non-linear plant, this approach is no longer optimal for the intermediate wind speeds, in common with such conventional gain-scheduled controllers.

   • For the case where a gain-schedule is automatically tuned by IFT for varying wind speeds, the control is not optimal at any operating point, but it is globally optimal for a range of operating points. However, for the case (as with IFC), where the desired gain schedule is not linear, the controller may possibly behave poorly across the entire wind speed region. This case has more parallels with an LTI robust control design, which optimises globally but may be severely suboptimal for local operating points. An LPV robust control design may be superior in general, but accuracy of modelling is critical for such an approach.

This comparison of IFT control with robust control will be included in the conclusions. Finally, the authors would like to state that the IFT approach in this paper, while interesting and novel, definitely stands to be further improved, and forms part of our future work.

19. Chapter 6 deals with results. Results are good.
    We would like to thank the reviewer for their kind comments and constructive feedback.

---

## Author Response (AR1)

Date  August 31, 2016
Our reference  n/a
Contact person  S. T. Navalkar
Telephone  +31 (0)15 27 83519
E-mail  S.T.Navalkar@TUDelft.nl
Subject  Response to reviewers

**Delft University of Technology**

Delft Center for Systems and Control

Address
Mekelweg 2 (3ME building)
2628 CD Delft
The Netherlands

www.dcsc.tudelft.nl

To
The Reviewers
*Wind Energy Science*

Dear Reviewer,

First of all, the authors would like to thank the reviewers for their positive and constructive feedback. We believe that the comments have helped us improve the quality of the paper. In our attempt to account for the comments, we have revises different aspects of the paper. The objective of this document is to respond to the points raised by the reviewers and to provide a detailed overview of the changes made to the paper.

Yours sincerely,

Sachin T. Navalkar

Enclosure(s):  Response to comments of Reviewer 1
Response to comments of Reviewer 2

**Response to comments of Reviewer 1**

1. Page 1, line 7. "The inertia of the flaps was tuned..." Nowhere in the paper the authors present this inertia tuning. Please comment on this.
   **Page 9, Lines 12-16, Page 10, Figure 15:** During the design of the experiment, the flutter analysis conducted on the numerical model of the blade showed high sensitivity to the exact value of the flap inertia about its free hinge axis. As such, a sensitivity analysis was conducted to ensure that the blade does not enter the flutter regime prematurely, and that the system is stable in the intended regime of operation. This sensitivity analysis had been omitted from the paper, but, as per the feedback of the reviewer, has been included in the revised version of the manuscript.

2. Section 2. The authors write they have scaled the INNWIND.EU rotor, but the scaling laws are not reported (time ratio, length ratio, Lock, Reynolds, etc.). Moreover, the scaled model is a 2-bladed rotor, the INNWIND.EU a 3-bladed. The only information provided is (page 3, line 24-25) is on the first blade freq. wrt the 1P (¿3.5) which is a standard value for all 3-bladed rotor (to avoid intersections in the Campbell diagram between the blades and the 1/2/3P). It looks like the scaled model is a proof of concept of technology more than a scaled model of a full scale wind turbine, so that the table 1 and the reference to the INNWIND rotor should be removed. Otherwise the authors must give more info about the scaling.
   The authors agree with the reviewer that the description of the scaling, especially with relation to the INNWIND.EU rotor, is inadequate. However, due to practical considerations, scaling was only done to ensure that the frequency ratio of the first blade mode with respect to 1P remained constant. As such, the authors agree that this scaled turbine simply represents a model for the proof of concept, and the reference to the INNWIND.EU rotor has been removed.

3. Section 2.1. No information is provided about the aerodynamic design: how the authors have chosen airfoils, chord and twist distribution. May the authors include some more info about this and some more data about the overall performances (power coefficients VS TSR, for instance)?
   **Page 3, Lines 12-13:** The reviewer raises an important point, it needs to be mentioned that the aerodynamic design of this specific rotor follows the design approach of Van Wingerden et al. (2011), where a similar rotor, but with conventional flaps (and no pitch control) was tested under similar experimental conditions. The authors of the current paper now provide the necessary reference, "Design of a scaled wind turbine with a smart rotor for dynamic load control experiments" by Hulskamp et al, 2011, and related, where a detailed analysis of the aerodynamic design and other performance characteristics can be found.

4. Page 5, lines 5-10. The authors present a mismatch between the measured and calculated structural behavior. The reason of this has been identified in the anisotropic behavior of the real blade not modeled in the isotropic FEM model. The authors should comment why they have not tried to identified this anisotropic behavior on some specimens (as done for Figure 1...) and then used an anisotropic FEM model.
   **Page 6, Figure 5:** As noted, there was indeed a difference between the measured and calculated structural behaviour. The manufacturer-published values of anisotropic material properties were used to estimate the stiffness of the blade. However, it was observed that the stiffness of the blade is not significantly affected by using an anisotropic model. The authors retested the stiffness of the blade, and it was found that the clamp used for fixing the blade root was not ideal, but allowed rigid-body rotation of the blade. The rotation of the blade root with increasing load was measured, and compensated for in order to correctly estimate the actual stiffness of the blade. The corrected figure has been used in the revised manuscript.

5. Page 7, line 10. "..adds an additional rigid-body degree of freedom": this comment is unnecessary because this is well-known; moreover this comment does not need to be supported by two (auto)citations.
   The authors agree that the statement is self-evident, and has been removed.

6. Page 7, lines 18-22. The discrepancy (about 20%) between the two mathematical models is an error in the modeling: a correct FEM model and a correct cross-sectional code + beam model can give the same (correct) results. In a journal paper this should be correct. Moreover, both the models return huge error wrt the real one (see previous point...).
   **Page 7, Lines 18-21, Page 8, Lines 1-6:** Indeed, there was a large difference in the modal description of the two mathematical models. This was a result of a difference in the clamping conditions: while the blade root section was completely clamped in the NASTRAN model, only the connecting nut was clamped in the Solidworks model. When this discrepancy was removed by correctly clamping the Solidworks model, the difference between the two models reduces to a large extent; the recalculated difference being less than 2.5%.

   - 1st Flapwise frequency: 18.97 Hz (Solidworks), 19.44 Hz (NASTRAN)
   - 1st Edgewise frequency: 78.37 Hz (Solidworks), 76.67 Hz (NASTRAN)
   - 2nd Flapwise frequency: 84.8 Hz (Solidworks), 87.88 Hz (NASTRAN)

The revised manuscript contains this updated modal analysis. As far as the difference between the numerical and experimental model is concerned, it should be noted that the numerically calculated values are blade frequencies; the rotor modal frequencies measured in practice are also influenced by the blade connection flexibility, motor stiffness and hub flexibility, and as such are necessarily lower than the numerically calculated blade modal frequencies. The difference, or uncertainty, in rotor modal frequency estimation is a powerful motivation for the choice of a self-tuning regulator as the load controller, since it can adapt to the true system parameters automatically. This explanation has been included in the manuscript.

7. Page 9, lines 1-2. This is not clear. The airspeed 36m/s refers to the scaled or the full scale model (it looks the full one...)? 340rpm is the scaled one. Probably the authors should present a regulation trajectory of the (scaled) wind turbine (i.e. rotor speed VS wind).

   **Page 8, Lines 13-16:** The terminology "air speed" did appear unclear, it is the resultant air speed incident on the blade, which is a combination of the inflow wind speed and the wind speed induced by the rotation of the blade. This definition has been made clear in the revised manuscript. The regulation trajectory of the scaled wind turbine is linear, and as such can be described by a single coefficient denoting its slope, which is 51.1 rpm/(m/s). This value has also been stated clearly in the revised manuscript.

8. Page 9, figures 13-14. The flutter analysis presented here looks more the one used for fixed wing (i.e. uniform airflow on the blade, constant AoA, no rotation). Is it also applicable on a rotation blade? Please add some comments in the paper about this flutter analysis. The wind speed on the x-axis refers to the scaled model?

   **Page 9, Lines 1-7:** As pointed out by the reviewer, the flutter analysis presented is indeed carried out for a fixed wing, with uniform airflow and constant angle of attack. The effect of rotation is only included in the sense that the incident wind speed is not the inflow wind speed, but rather, the total air speed as defined in the previous comment. This forms a first approximation to the true flutter behaviour of the rotating blade, the inflow conditions along the span of the blade are held constant to those obtained at the tip section. While the error in the aerodynamic forces increases for the sections radially inboard, these sections also undergo smaller structural motions and hence contribute to a progressively smaller extent to the aeroelastic behaviour and hence the modal analysis of the blade as a whole. Further, the blade is typically twisted such that the angle of attack remains constant throughout the blade span; this is approximated by using aerodynamic panels with no twist, and with a constant angle of attack, along the span of the blade. As such, it is assumed that the flutter analysis of the non-rotating blade is a good approximation of the dynamic behaviour of the actual experimental blade. This explanation has been included in the revised manuscript. Finally, it should be pointed out that the "air speed" in the Figures 13 and 14 refers to the total air speed, as defined in the previous comment; this has also been clarified.

9. Page 16, lines 4-7. Again more information about the operation of the model is necessary: if the rated speed is 4.5m/s at 230rpm, at 6m/s the rotor speed of a classical variable-speed pitch-regulated wind turbine is again 230rpm (i.e. in the above-rated region the rotor speed is kept constant). The authors must better define the regulation of the model.

   **Page 11, Lines 4-8:** Thank you for this remark. As described in the second paragraph of Section 4, the load resistance connected to the wind turbine generator is kept constant; this constant load operation implies that the rotor speed increasely linearly with the wind speed at a rate of 51.1 rpm/(m/s). This is indeed not similar to the operation of a classical variable-speed pitch regulated wind turbine, where collective pitch control is used to ensure that the rotor speed remains constant (above-rated) irrespective of the change in wind speed. However, the constant resistance operation of the turbine serves three main purposes: it describes the effect of (temporary) overspeeds which may cause the turbine to enter flutter, it emulates the behaviour of the turbine under below-rated conditions, and it describes the potential of an adaptive control strategy that may be required to retune itself under varying operating conditions. This description and motivation of constant load operation has been clarified in the revised manuscript.

10. Some figures may be more readable if different line styles are used (i.e. solid, dash-dotted, dotted, etc..). This helps if read on black/white copies or by color-blind person. Page 6, line 19. Add extra space: Finally,an.

    The figures and the typo have been revised as advised.

11. Page 8, fig 9-10. Please check these figures, because they look inverted (Fig. 9 looks the first edgewise mode...).

    **Page 8, Figures 9-10:** The figures were checked and found to be correct. The flapwise modes display deformation perpendicular to the aerodynamic panels, while the lead-lag mode displays deformation in the plane of the aerodynamic panels. This description has been updated in the revised manuscript.

12. Page 12, line 7, correct "currrent". Sections 6.2, 6.3, 6.4. In the titles the word tuning should be removed since already included in the acronym "IFT". Figures 21-24: why the words PRE-POST FLUTTER are uppercase?

    The corrections have been made as advised by the reviewer, we thank them once again for their constructive feedback.

**Response to comments of Reviewer 2**

1. Please precise what is new, the combination of the 2 control strategy? The real time aspect? Referring to your personal bibliography some of these aspects has already been treated. The reader should clearly be able to locate this new article in your scope.

   **Page 3, Lines 19-26:** Thank you for this remark, the original introduction was indeed unclear regarding the precise aspect in which the paper represents an advancement of the state of the art. To answer the question, three main novelties can be found in this paper:

   (a) This is the first experimental demonstration of combined pitch and flap control.

   (b) This is the first experimental demonstration of free-floating flaps applied to rotating wind turbine blades. This is also the first time their potential has been demonstrated for load reduction in wind turbines experimentally. Further, this is the first time that free-floating flaps have been shown to induce flutter on wind turbine blades experimentally.

   (c) This is the first experiment where IFT has been devised and implemented for adaptively tuning the gain schedule of a nonlinear (LPV) system.

   Since these novelties were not clear, they have been mentioned explicitly in the introduction of the revised manuscript.

2. Line 3 page 2: please defined 1P in this chapter, the definition of this acronym comes in chapter 2 (too late),

   **Page 1, Line 16:** The term 1P has been defined as the rotor speed in the introduction of the revised version of the manuscript.

3. You affirmed that "Data-driven controller may be able to achieve greater optimality of performance without the excessive conservatism of the true robust design"; It seems you want to compare totally opposite methodology. Data-driven has no proof of robustness and/or optimality and the performance of the controller depend of the number of data-acquisition set you used to determine your controller. This methodology has advantages of deleted identification part and to be simple to use and implemented but does your non-linear system is not a linear piecewise system? Does you have some robust control methodology that you could use?

   **Page 2, Lines 34-35, Page 3, Lines 1-4:** The authors agree completely with the reviewer that the data-driven strategy described in this paper has no proof of robustness or optimality, and depends strongly on the realisations of the data and wind speed during operation. In case one were to approach the controller design from a traditional robust mindset, one would be required to formulate an accurate linear parameter-varying or piece-wise linear description of the system, which would undoubtedly be subject to significant parametric and dynamic uncertainties. Formulating a robust controller based on such an uncertain plant description could yield a strongly conservative controller that is perhaps not able to achieve the maximum possible load reductions. The advantage of the IFT approach is that, besides being simple, it uses input-output data to tune itself to optimise a simple user-defined criterion. Since a feedforward approach is used, the IFT controller cannot destabilise the plant; in the best case, it could perhaps achieve load reductions that a conservative robust controller may not be able to reach.
   This discussion has been repeated in the revised manuscript to clearly compare the proposed control strategy with the robust control approach.

4. You must prcised clearly what is new compared to your bibliography. The combination of the methodology? The real time aspect? If we read some paper of your bibliography it seems that all aspect is already treated in your previous article.
   Please refer to the discussion related to the first comment.

5. This chapter must be design to offer real investigation and comprehension of the design and manufacturing of blades for people that cannot be accustomed to these topics. Why this parts is important to understand the following parts of this article?

   **Page 4, Lines 2-5:** The authors apologise for the lack of clarity in describing the aim of Section 2. In principle, this section describes the design of the experimental setup, and provides details regarding the materials, method of manufacture and assembly. The authors believe that this is important, since it is the first time that a wind turbine blade has been manufactured to incorporate free-floating flaps. Primarily, the destabilising effect of the free-floating flap is studied in detail, and the parameters are tuned such that the blade is close to, but not beyond, the flutter point in order that maximal control authority is achieved. This reasoning has been included in the introductory paragraph of Section 2.

6. Figure 3 is cited befor figure 1 and 2.
   This has been rectified by reordering the figures.

7. Line 8 page 4 : what does you mean with this word "qua"?
   "Qua" has the meaning "in terms of".

8. Line 24 page 4 :what does you mean with CAD?
   **Page 5, Line 17:** CAD stands for Computer-Aided Design, the expansion of the acronym has been included in the revised manuscript.

9. Figure 2 : can you explain where is the sample and what is around the sample?
   **Page 5, Figure 4:** The top grey rectangle and the two black rectangles below it are the 3d-printed plastic samples, bonded with carbon fibre. They are placed on a sandstone-coloured desktop, this forms the background. This will be made clear in the caption.

10. You found that the blade designed using CAD software is 30% stiffer than the actual manufactured blade but doesnt explain why. Please avoided postulated affirmation and concretely explained the difference and the consequence on the announced gain of 70%. 30 % more stiffness of the blades does not seem to be reflected in the same way on the resonance frequency that not only down 13%. Can you explain why?
    **Page 5, Figure 3, Page 8, Lines 2-6:** The authors retested the stiffness of the blade, and it was found that the clamp used for fixing the blade root was not ideal, but allowed rigid-body rotation of the blade. The rotation of the blade root with increasing load was measured, and compensated for in order to correctly estimate the actual stiffness of the blade. The corrected figure has been used in the revised manuscript. The announced gain of 70% is an experimentally measured quantity, and has no relation to the numerical models (that do not predict load-reduction capabilities). As far as the difference between the numerical and experimental model is concerned, it should be noted that the numerically calculated values are blade frequencies; the rotor modal frequencies measured in practice are also influenced by the blade connection flexibility, motor stiffness and hub flexibility, and as such are necessarily lower than the numerically calculated blade modal frequencies. These issues have been clarified in the revised manuscript.

11. Line 14 page 11 you talk about an optimal control action, where is the proof of the optimality of the control action? Do you have references that assess that the control action is optimal? Its an global optimality or an operating point optimal controller?

    **Page 12, Lines 3-7:** The reviewer points out an important issue: IFT has no theoretical proof of global optimality. Indeed, if IFT is used for tuning a feedback controller, there is no guarantee that the closed-loop system will be stable. Since the paper discusses a feedforward controller, this is not an issue of concern. Further, if the step-size in the gradient descent algorithms is too large, the parameter tuning process may become unstable. These issues have been dealt with by Hjalmarsson in the reference "Iterative feedback tuning: an overview." This paper has been referenced in Section 5 in the revised manuscript. The "optimality" described in this section refers specifically to the optimisation in a local sense, of the user-defined cost function. This has also been made explicitly clear in the revised text.

12. Line 19 page 11, Does the plant or the controller is assumed be still LPV?

    **Page 12, Lines 12-13:** The original statement is ambiguous. The plant, being the wind turbine, is at all times LPV. For the case where the wind conditions are held constant in the wind tunnel, an LTI controller is tuned by IFT for that specific operating point of the LPV plant. On the other hand, for the case where the wind conditions are allowed to vary in the wind tunnel, a gain-scheduled controller is tuned by the IFT algorithm developed in this paper. This schema has been explicitly mentioned in the introduction of Section 5.

13. Line 20 page 11: What is the consequence of the assumption of constant wind speed during each set of IFT experiments?

    **Page 12, Lines 14-15, Page 13, Lines 1-2:** The assumption of constant wind speed implies that ordinary IFT can yield an optimal load-reducing controller only for that specific operating point. Such a controller may not achieve the highest possible load reductions, or may even increase loads, at other operating points. It is for this reason that the ordinary IFT process has to be repeated for different constant wind speeds, or an IFT gain schedule has to be generated for a varying wind speed. This has been made clear in the revised manuscript.

14. Line 32 page 11 you wrote that Ak, Bk, Ck, Dk are considered unknown so how you assess that this same matrices could be written as equation 3. What does the bracket [0] or [1] means?

    **Page 13, Lines 14-20:** $A_k$, $B_k$, $C_k$ and $D_k$ are all considered unknown but assumed to admit a specific LPV structure, often used for modelling wind turbines, as in the PhD thesis of Van Wingerden; a reference to this thesis will be provided in the revised text. The terms $A^{[0]}$ and $A^{[1]}$ signify the unknown components of $A_k$, one which is constant over time, and one that varies linearly with the wind speed. As per Van Wingerden 2008, there should be one more term that varies with $V_k^2$, however the influence of this term is small and it is neglected in this paper. This description has been added to the revised text.

15. Equation 5 page 12, you use qk and write line 22 page 12 that it will be described in the next section but in section 5.2 you wrote that qk is equal to zero. What the interest of this variable? This variable doesnt appear on the block diagram figure 16.

    **Page 14, Figure 17:** The authors thank the reviewer for pointing out this issue. While $q_k$ is indeed zero for the reference experiment, it is non-zero for the gradient experiment of IFT, please see equation (11). As such, it is important for determining the gradient of the performance criterion with respect to the controller parameters. The authors have update Figure 17 such that this signal is included in the block diagram.

16. Where is the global control scheme (IPC+IFC+SYSTEMS+PID...)?

    **Page 14, Figure 17:** The block diagram in Figure 17 has been extended with the stabilising collocated PID control.

17. Line 9 page 14: $\mu$ is not defined (a residual of Sachin T. Navalkar et al. / IFAC-PapersOnLine 48-26 (2015)?)

    **Page 15, Lines 23:** The authors thank the reviewer for pointing out this typo, $\mu_*$ has been replaced by $V_*$.

18. 1st paragraph Page 15 : you precise that your controller is optimal for the operating point that assess you cant compared it to robust control. What happen when the wind speed is between two operating point you use to find your controller?

    **Page 23, Lines 4-14:** The reviewer points out a shortcoming of the two control approaches used in the paper:

    - For the case where LTI controllers are devised for constant operating points, the control action for an intermediate speed is obtained by interpolating between the gains of the closest wind speeds. This is the approach followed by most industrial gain-scheduled controllers. For a highly non-linear plant, this approach is no longer optimal for the intermediate wind speeds, in common with such conventional gain-scheduled controllers.

- For the case where a gain-schedule is automatically tuned by IFT for varying wind speeds, the control is not optimal at any operating point, but it is globally optimal for a range of operating points. However, for the case (as with IFC), where the desired gain schedule is not linear, the controller may possibly behave poorly across the entire wind speed region. This case has more parallels with an LTI robust control design, which optimises globally but may be severely suboptimal for local operating points. An LPV robust control design may be superior in general, but accuracy of modelling is critical for such an approach.

This comparison of IFT control with robust control has been included in the conclusions. Finally, the authors would like to state that the IFT approach in this paper, while interesting and novel, definitely stands to be further improved, and forms part of our future work.

19. Chapter 6 deals with results. Results are good.
    We would like to thank the reviewer for their kind comments and constructive feedback.

---

## Author Response (AR2)

| | |
|---|---|
| Date | September 14, 2016 |
| Our reference | n/a |
| Contact person | S. T. Navalkar |
| Telephone | +31 (0)15 27 83519 |
| E-mail | S.T.Navalkar@TUDelft.nl |
| Subject | Response to editors |

**Delft University of Technology**

Delft Center for Systems and Control

Address
Mekelweg 2 (3ME building)
2628 CD Delft
The Netherlands

www.dcsc.tudelft.nl

To
The Editors
*Wind Energy Science*

Dear Prof. Aubrun and Prof. Mann,

First of all, on behalf of the co-authors, I would like to thank the editors for their positive and constructive feedback. We believe that the comments have helped us improve the quality of the paper. In our attempt to account for the comments, we have removed the redundant figures from the manuscript.

We hope that, with this amendment, the manuscript is ready for publication in the journal.

Yours sincerely,

Sachin T. Navalkar

Enclosure(s): Revised Manuscript